# Cryo-EM structure of a functional monomeric Photosystem I from *Thermosynechococcus elongatus* reveals red chlorophyll cluster

Orkun Çoruh [1,2,9], Anna Frank [3,9], Hideaki Tanaka[1], Akihiro Kawamoto [1], Eithar El-Mohsnawy[4], Takayuki Kato [5], Keiichi Namba [6,7,8], Christoph Gerle [1✉], Marc M. Nowaczyk [3✉] & Genji Kurisu [1,2✉]

A high-resolution structure of trimeric cyanobacterial Photosystem I (PSI) from *Thermosynechococcus elongatus* was reported as the first atomic model of PSI almost 20 years ago. However, the monomeric PSI structure has not yet been reported despite long-standing interest in its structure and extensive spectroscopic characterization of the loss of red chlorophylls upon monomerization. Here, we describe the structure of monomeric PSI from *Thermosynechococcus elongatus* BP-1. Comparison with the trimer structure gave detailed insights into monomerization-induced changes in both the central trimerization domain and the peripheral regions of the complex. Monomerization-induced loss of red chlorophylls is assigned to a cluster of chlorophylls adjacent to PsaX. Based on our findings, we propose a role of PsaX in the stabilization of red chlorophylls and that lipids of the surrounding membrane present a major source of thermal energy for uphill excitation energy transfer from red chlorophylls to P700.

[1] Laboratory for Protein Crystallography, Institute for Protein Research, Osaka University, Suita, Osaka, Japan. [2] Department of Macromolecular Science, Graduate School of Science, Osaka University, Toyonaka, Osaka, Japan. [3] Plant Biochemistry, Faculty of Biology and Biotechnology, Ruhr-University Bochum, Bochum, Germany. [4] Department of Botany and Microbiology, Faculty of Science, Kafrelsheikh University, Kafr Al Sheikh, Egypt. [5] Laboratory of CryoEM Structural Biology, Institute for Protein Research, Osaka University, Suita, Osaka, Japan. [6] Graduate School of Frontier Biosciences, Osaka University, Suita, Osaka, Japan. [7] RIKEN Center for Biosystems Dynamics Research and SPring-8 Center, Suita, Osaka, Japan. [8] JEOL YOKOGUSHI Research Alliance Laboratories, Osaka University, Suita, Osaka, Japan. [9] These authors contributed equally: Orkun Çoruh, Anna Frank. ✉email: gerle.christoph@protein.osaka-u.ac.jp; marc.m.nowaczyk@rub.de; gkurisu@protein.osaka-u.ac.jp

Oxygenic photosynthesis by plants, algae, and cyanobacteria produces almost all reducing equivalents of our biosphere and all the oxygen needed to power life[1]. The light reactions of photosynthesis are executed by two large pigment-binding membrane protein complexes termed Photosystem I (PSI) and Photosystem II (PSII)[2]. PSI is a light-driven oxido-reductase, which accepts electrons via water-soluble plastocyanin or cytochrome $c_6$ on the lumenal side of the thylakoid membrane and donates electrons to ferredoxin or flavodoxin on its stromal side mainly for the reduction of $NADP^+$ to NADPH by FNR, for fixation of $CO_2$ availing the reductive pentose phosphate cycle[3–6].

Though cyanobacterial PSI exists in monomeric and dimeric form in the native membrane, its dominant oligomeric form is a trimer[7–10]. Pioneering work on the isolation and characterization of trimeric, dimeric, and monomeric PSI from *T. elongatus* established the PSI monomer as the minimal functional unit of cyanobacterial PSI and described its overall shape by negative stain electron microscopy (EM) and image analysis[10]. Cyanobacterial PSI is not necessarily bound to membrane-intrinsic light-harvesting complexes while strictly monomeric PSI of algae and plants forms supercomplexes with varying numbers of light-harvesting complexes that act as an external antenna[5,11]. The striking difference in oligomeric states and mode of light harvesting has been the framework of several molecular scenarios in the evolution of photosynthesis from its beginning in cyanobacteria ~3 billion years ago, over algae to the emergence of higher plants on land[12,13]. A high-resolution X-ray crystal structure of the trimeric PSI from *Thermosynechococcus elongatus* (hereafter referred to as trimer crystal structure) was reported almost 20 years ago at 2.5 Å resolution[8], providing a detailed description of almost all of its protein components and their bound ligands. The PSI trimer has a total molecular mass of 1080 kDa with the monomer comprising the 12 protein subunits, PsaA, PsaB, PsaC, PsaD, PsaE, PsaF, PsaI, PsaJ, PsaK, PsaL, PsaM, PsaX, 96 chlorophylls, more than 20 carotenoids, four structural lipids, and three $[Fe_4S_4]$ iron–sulfur clusters. The chlorophyll content consists of two functionally distinct groups. The first group is composed of six chlorophylls of the electron transfer chain (ETC) including the special pair chlorophylls Chl*a* and Chl*a*′ (P700) and is involved in electron pumping. The second group accommodates the antenna chlorophylls involved in light harvesting and excitation energy transfer. Antenna chlorophylls can be further divided into the chlorophylls of the core and the peripheral antenna which are organized into lumenal and stromal side networks. In contrast to the chlorophylls, the carotenoids bound to PSI are excluded from the core of the complex and located in the peripheral antenna where they span the transmembrane region between lumenal and stromal sides.

There has been a long-standing interest in the isolation, biochemical characterization, spectroscopic analysis, and high-resolution structure determination of monomeric cyanobacterial PSI. However, to date, with a notable exception of the PSI monomer structure from *Synechocystis* sp. PCC 6803 solved at 4.0 Å resolution by X-ray crystallography[14], all structures of cyanobacterial PSI have been determined from trimeric complexes[8,15–21] or the recently discovered tetrameric PSI from *Anabaena* sp. PCC 7120[22–24] and *Chroococcidiopsis* sp. TS-821[25]. This indicates possible challenges in both the purification of monomeric PSI and the growth of well-ordered 3D crystals for high-resolution structure determination by X-ray crystallography. Meanwhile, the ongoing 'Resolution Revolution' in cryo-electron microscopy (cryo-EM) structural analysis[26] has made structure determination of large integral membrane protein complexes feasible in the absence of crystals. Furthermore, the ability of cryo-EM single particle analysis to solve structures of soluble and

membrane proteins using relatively inexpensive 200 kV cryo electron microscopes is now well established[27,28]. A detailed structural analysis of monomeric PSI may allow a better understanding of the mechanism and biological meaning of PSI oligomerization. Another interest in monomeric cyanobacterial PSI originates from the spectroscopically discovered loss of long-wavelength chlorophylls, also named red chlorophylls, from trimeric PSI upon monomerization[29–31]. Red chlorophylls allow uphill excitation energy transfer to P700 for its oxidation via red light beyond 700 nm by using thermal energy supplied by the vibrations of the surrounding phonon bath, effectively expanding the light spectrum available for electron pumping[32–34]. Thus far spectroscopy performed at room temperature identified three distinct absorption bands for long-wavelength chlorophylls at 708 nm, 715 nm, and 719 nm[35–37]. In some cyanobacteria, e.g. *Halomicronema hongdechloris*, red chlorophylls are specialized as chlorophyll *f*[38], which in principle can be identified in high-resolution structures[17,18,39]. However, PSI of *T. elongatus* contains only chlorophyll *a*, whose absorption spectra are tuned by their local chemical environment. Thus, direct structural identification of the red chlorophylls in PSI of *T. elongatus* is not possible.

Here, we describe a mild purification method for high-yield isolation, spectroscopic characterization, and structure determination of monomeric PSI from *T. elongatus*. After confirming the complex's completeness and activity, we determined its structure to 6.5 Å resolution using X-ray crystallography and to 3.2 Å using single particle cryo-EM with image data collected on a CRYO ARM 200 (JEOL) operated at 200 kV. Our analyses provide detailed structural insight into the flexibility introduced by the monomerization process and allow us to identify membrane-facing clusters of lost red chlorophylls.

## Results

**Monomer purification and characterization by mass spectrometry.** We successfully isolated functional PSI monomers in large quantities by solubilization of the membranes with lauryldimethylamine oxide (LDAO) and purification via hydrophobic interaction chromatography (HIC) (Fig. 1 and Supplementary Fig. 1). Single bands ~1000 kDa (PSI trimer) and ~330 kDa (PSI monomer) on blue native gels (Fig. 1a) indicate high sample purity. SDS-PAGE analysis (Fig. 1b) together with two-dimensional polyacrylamide gel electrophoresis (2D-PAGE; Fig. 1c) revealed that all major subunits of the complex were preserved during purification. The subunit composition of both trimeric and monomeric PSI species was further investigated by Matrix-assisted laser desorption/ionization time-of-flight mass-spectrometry (MALDI-TOF MS) (Fig. 1d, e). The peaks determined by MALDI-TOF MS analysis accurately correspond to the masses of the PSI subunits known from the published high-resolution X-ray crystal structure[8]. Minor differences were interpreted as post-translational modifications. All theoretically calculated and practically determined masses are summarized in Table 1. The results confirmed the presence of all PSI subunits except for PsaA and PsaB, which are not in the detection range of the MALDI-TOF instrument. However, these subunits were resolved by SDS-PAGE analysis (Fig. 1b). Particularly, the presence of PsaK and PsaL, which are easily lost, confirms gentle purification of the complex. The functionality of purified PSI was validated by the measured activities, which are $1970 \pm 306$ µmol $O_2 \cdot$ mg Chl$^{-1} \cdot$ ml$^{-1} \cdot$ h$^{-1}$ for the monomeric and $1793 \pm 211$ µmol $O_2 \cdot$ mg Chl$^{-1} \cdot$ ml$^{-1} \cdot$ h$^{-1}$ for the trimeric complex. The quality of isolated monomeric PSI was further confirmed by negative stain EM observation, demonstrating the monodispersity of the protein complexes in solution as well as homogeneity of the

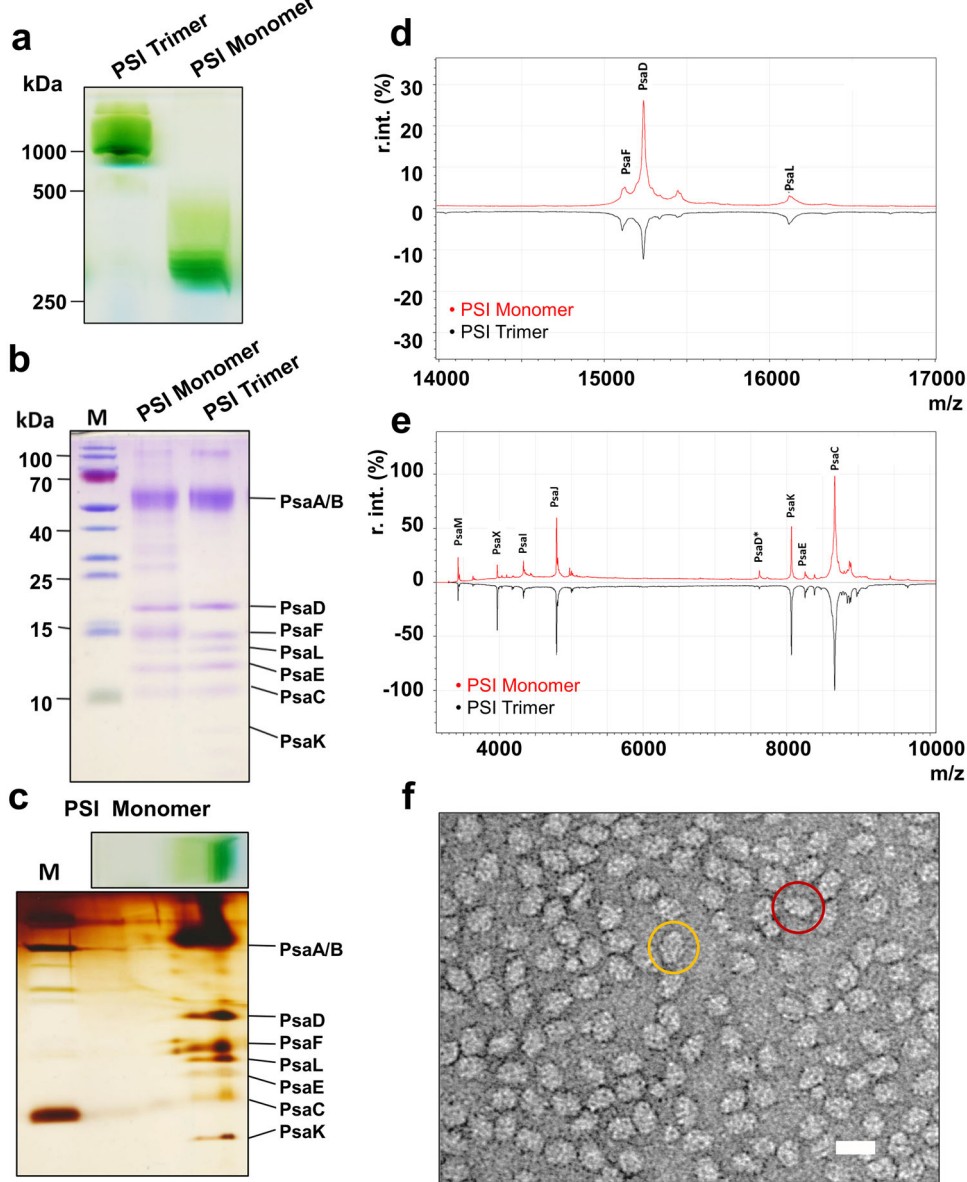

**Fig. 1 Biochemical characterization of the purified PSI monomer complex from *T. elongatus*. a** Blue-native (BN) PAGE of the PSI monomer next to a PSI trimer control (3.5 µg Chl/lane). **b** SDS-PAGE of monomeric and trimeric PSI in comparison (2.5 µg Chl/lane) **c** Two-dimensional (2D)-PAGE of monomeric PSI (3.5 µg Chl/BN-lane). **d**, **e** MALDI-TO Fmass-spectrometry of monomeric and trimeric PSI complexes, mass range: **d** 14,000–17,000 *m/z* and **e** 3000–10,000 *m/z* (see also Table 1), *z=2. **f** Negative stain EM image of PSI monomer with a top view encircled in yellow and a side view encircled in red. Scale bar corresponds to 20 nm.

overall sample (Fig. 1f). The particle size and shape were consistent with previous work[10].

**UV-Vis spectroscopy.** UV-Vis spectra of trimeric and monomeric PSI were measured at room temperature (Fig. 2a) and difference spectra (trimer–monomer) were calculated after normalization at the Chlorophyll a peak (678 nm; Fig. 2b). Positive peaks between 400 and 500 nm imply a higher amount of carotenoids in trimeric PSI. Additionally, a shift in the absorption of trimeric PSI towards longer wavelengths is detectable around 700–725 nm. This shift indicates a higher amount of red chlorophylls in the PSI trimers and is in accordance with previous results[40,41].

Fluorescence emission spectra of both trimeric and monomeric PSI were measured at 77 K (Fig. 2c). Characteristic peaks ~730 nm (trimer) and 720 nm (monomer) were observed. This blue

shift of chlorophyll fluorescence confirms the loss of red chlorophylls in PSI monomers. Moreover, the spectra might indicate the presence of uncoupled chlorophylls (characteristic fluorescence at 689 nm) in the monomer preparation.

**Structure determination.** We used single particle cryo-EM and X-ray crystallography to determine the structure of monomeric PSI from *T. elongatus* at 3.2 Å (PDB ID 6LU1, EMDB ID EMD-0977) and 6.5 Å (PDB ID 7WB2; Supplementary Fig. 2), respectively.

For single particle cryo-EM, PSI monomer purification was optimized by exchanging the detergent from *n*-dodecyl β-D-maltoside (β-DDM) to glyco-diosgenin (GDN)[42] and removal of excess free detergent using the GraDeR approach[43]. PSI monomer particles were packed tightly to the edge of holes of the holey carbon film and had a strong tendency of preferred

**Table 1 Calculated and determined masses of PSI subunits from *T. elongatus*.**

| Subunit | Calculated mass (Da) | Determined mass (Da) | Difference | Modification |
|---|---|---|---|---|
| PsaA | 84,872 | Not detected | – | – |
| PsaB | 83,044 | Not detected | – | – |
| PsaC | 8800.1 | 8667.37 | −132.73 | N-methionine deleted |
| PsaD | 15370.5 | 15237.49 | −133.01 | N-methionine deleted |
| PsaD* | 7685.25 | 7618.86 | −66.39 | N-methionine deleted |
| PsaE | 9388.5 | 8255.43 | −133.07 | N-methionine deleted |
| PsaF | 15113.4 | 15108.52 | −4.88 | - |
| PsaI | 4166 | 4332.97 | +35.97 | Acetyl group added |
| PsaJ | 4755.7 | 4793.64 | +26.94 | Formyl group added |
| PsaK | 8480 | 8065.36 | −414.64 | -(N-MVLA) |
| PsaL | 16251 | 16.114.90 | −131.10 | N-methionine deleted |
| PsaM | 3424.1 | 3423.10 | −1.00 | – |
| PsaX | 4100.9 | 3968.79 | −132.11 | N-methionine deleted |

All data calculated via https://web.expasy.org/protparam/.
*z=2.

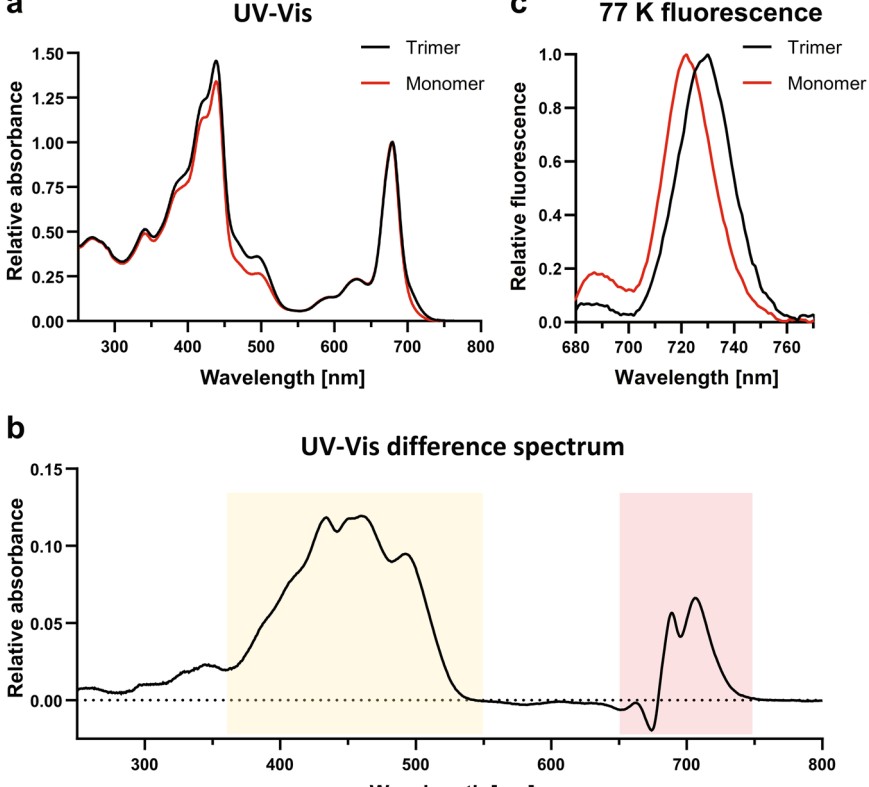

**Fig. 2 UV-Vis and 77 K spectra of trimeric and monomeric PSI from *T. elongatus*. a** UV-Vis spectra. GraDeR[43]-prepared samples were diluted in buffer B + 0.005% (w/v) GDN. **b** UV-Vis difference spectra (trimer–monomer) were calculated after normalization at the chlorophyll a peak (678 nm). Differences related to the absorption of carotenoids (orange) and red chlorophylls (red) are indicated by the colored background. **c** 77 K fluorescence spectra. Samples were diluted in buffer B + 0.03% (w/v) β-DDM. Measurements were carried out at an excitation wavelength of 436 nm with a step size of 1 nm and a bandpass filter of 5 nm.

orientation to show side views. The preferred side-view orientation, however, did not prohibit image analysis thanks to an even radial distribution of Euler angles (Supplementary Fig. 4). Refinement, Bayesian polishing, and CTF correction at the particle level in the process of image analysis using RELION 3.0[44] gave a final map at a resolution of 3.2 Å (Gold Standard FSC cutoff at 0.143, Supplementary Fig. 4) with a local resolution ranging from 2.75 Å at the core to 4.3 Å in the peripheral region (Fig. 3).

We employed the crystal structure model of *T. elongatus* trimeric PSI (PDB ID 1JB0)[8] as a starting model and were able to build all the models for expected subunits except PsaK and PsaX (see Table 2 for data collection and refinement statistics and Supplementary Table 1 for the percentage of modeled amino residues). X-ray diffraction data from the crystals were collected at the beamline BL44XU at SPring-8 up to 4.8 Å and were analyzed using a 6.5 Å cut-off. Our atomic model built from the

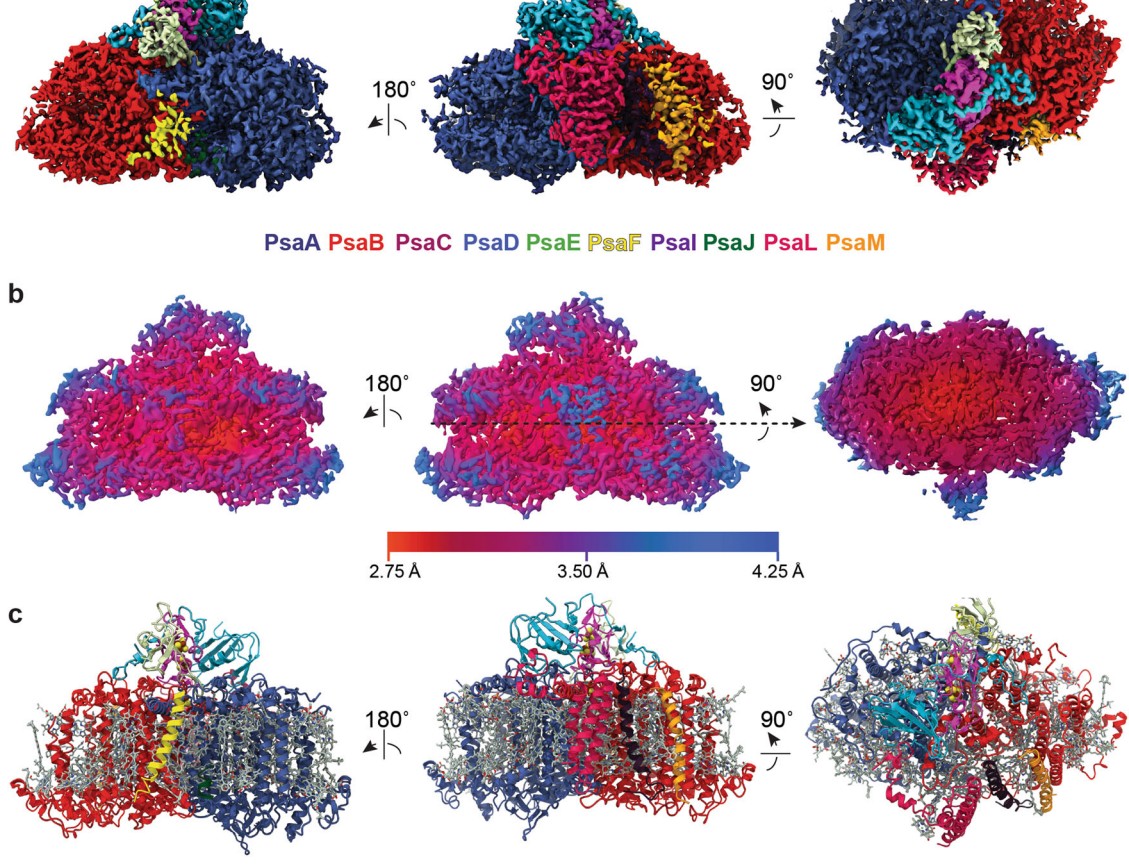

**Fig. 3 Overall structure of the PSI monomer from *T. elongatus*. a** The 3D cryo-EM Coulomb density map of the PSI monomer viewed parallel to membrane from the membrane side (left), trimerization side (middle), and viewed along the membrane normal from the stromal side. Map regions are colored according to the fitted model following the color scheme indicated by the subunit name font color. **b** Local resolution map of the PSI monomer viewed parallel to membrane from the membrane side (left), trimerization side (middle), and viewed along the membrane normal from the stromal side. Coloring by local resolution as indicated. **c** The structural model of the PSI monomer viewed parallel to membrane from the membrane side (left), trimerization side (middle), and viewed along the membrane normal from the stromal side. Coloring of subunits as in **a** and cofactors in gray.

X-ray diffraction data comprised all subunits including PsaK and PsaX (see Table 3 for data collection and refinement statistics).

**Overall architecture**. While cyanobacterial trimeric PSI resembles a clover leaf[8], the shape of monomeric PSI resembles a curling stone with the stromal subunits PsaC, PsaD, and PsaE forming the handle (Fig. 3a and Supplementary Movie 1). Our cryo-EM map was of sufficient quality to model amino acid side chains and also most of the chlorophyll phytol chains. Cofactor coordination via the protein environment is preserved between our monomeric cryo-EM structure and the trimeric crystal structure. The overall structure of monomeric PSI is very similar to that of trimeric PSI. However, differences were found within the peripheral subunits, including a changed conformation of the trimerization interface formed by the C-terminal region of PsaL, a disordered loop-helix-loop motif in PsaB (B291-B316), invisible PsaK and PsaX subunits and a changed occupation of peripheral chlorophylls and carotenoids. Our PSI monomer model contains 10 protein subunits featuring seven of the transmembrane subunits, PsaA, PsaB, PsaF, PsaI, PsaJ, PsaL, PsaM, and the three extrinsic stromal subunits PsaC, PsaD, PsaE and accommodates a total of 112 cofactors (Fig. 3c). Subunits and cofactors were found to be positioned akin to their trimeric counterparts retaining the same side chain coordination to the metals (Fig. 4). Newly built additions to the trimeric crystal structure-based model (PDB ID

1JB0) are residues A11-A12 and A262-A265 along with eight carotenoids.

**Cofactor structure**. Cofactors included in our model were built with confidence into well-defined regions of the Coulomb potential density map (>3 σ). In several of the modeled β-Carotenes, the visibility of methyl group bumps in the density map allowed us to discern the twist of the polyene chain (Fig. 4c and Supplementary Fig. 6). Among the chlorophyll-coordinating cofactors, we found a self-coordinating phytol chain (Fig. 4d), a phosphatidyl glycerol lipid (Fig. 4e), and a new β-carotene (BCR/B858). Positions of cofactors were mostly identical to the cofactors of the trimeric crystal structure; however, to fit them accurately into the cryo-EM density map, we modified some carotenoid and chlorophyll geometries such as the rotational angle of the carotenoid headgroup ring, chlorophyll chlorine ring orientation, and phytol chain length or direction. Cofactors of the ETC are well-resolved in our structure and, as expected, the positions of all six chlorophylls, two phylloquinones, and the three iron–sulfur clusters are identical to that of the trimeric crystal structure (Fig. 4a, b). The RMSZ values for bond lengths and bond angles define how the geometry of a modeled molecule satisfies the stereochemical restraints[45]. For the ETC chlorophylls in our model, these values lie between 0.84 and 0.95 for bond lengths and between 0.94 and 1.17 for bond angles, indicating the

**Table 2 Cryo-EM data collection, refinement and validation statistics.**

|  | PSI monomer<br>EMDB ID: EMD-0977<br>PDB ID: 6LU1<br>EMPIAR ID: 10352 |
|---|---|
| *Data collection and processing* |  |
| Microscope | CRYOARM 200 (JEOL) |
| Voltage (kV) | 200 |
| Camera | Gatan K2 Summit |
| Defocus range (μm) | 0.5–3.5 |
| Magnification | ×60,000 |
| Exposure time (s) | 12 |
| Number of frames per image | 60 |
| Electron exposure (frame/total) ($e^-/\text{Å}^2$) | 1.34/80.4 |
| Pixel size (Å) | 0.89 |
| Micrographs (initial/final) | 1530/1107 |
| Particles (initial/final) | 227,910/46,105 |
| Box Size (px) | 256 |
| Symmetry imposed | C1 |
| Map sharpening B-factor ($\text{Å}^2$) | −45.29 |
| FSC threshold/map resolution (Å) | 0.143/3.2 |
| *Refinement* |  |
| Initial model used (PDB code) | 1JB0 |
| Non-hydrogen atoms | 21,203 |
| Protein residues | 1986 |
| Ligands | 121 |
| R.M.S.D. (bonds (Å)/angles (°)) | 0.011/1.181 |
| Ramachandran favored (%) | 95 |
| Ramachandran allowed (%) | 5 |
| Ramachandran outliers (%) | 0 |
| Poor rotamers | 0.06 |
| Molprobity score | 1.49 |
| Clashscore | 4 |
| EMRinger score | 3.87 |

**Table 3 X-ray data collection and refinement statistics.**

|  | PSI monomer<br>PDB ID: 7WB2 |
|---|---|
| *Data collection* |  |
| X-ray source | SPring-8 BL44XU |
| Detector | MX-300HE |
| Wavelength (Å) | 0.90000 |
| Space group | $P\,3_2\,2\,1$ |
| Unit cell parameters |  |
| $a, b, c$ (Å) | 187.029, 187.029, 233.805 |
| $\beta$ (°) | 120 |
| Resolution range (Å) | 49.57–6.5 (6.732–6.5) |
| Total number of reflections | 102,562 (10,731) |
| Number of unique reflections | 9572 (949) |
| Multiplicity | 10.7 (11.3) |
| Completeness (%) | 98.46 (100) |
| $I/\sigma\,(I)$ | 29.58 (7.39) |
| $R$-merge ($I$) (%) | 0.05486 (0.3354) |
| CC½ (%) | 0.999 (0.978) |
| *Refinement* |  |
| Resolution range (Å) | 49.57–6.5 |
| No. reflections | 9556 |
| $R_{\text{work}}$ (%) | 0.3940 |
| $R_{\text{free}}$ (%) | 0.4616 |
| Total number of atoms | 23,837 |
| Averaged B-factor ($\text{Å}^2$) | 259.26 |
| R.M.S.D. (bonds (Å)/angles (o)) | 0.033/2.11 |
| Ramachandran favored (%) | 93.86 |
| Ramachandran allowed (%) | 5.69 |
| Ramachandran outliers (%) | 0.55 |

The numerals in parentheses are for the highest resolution shell.

soundness of our model. Distances between the cofactors of both ETC branches as measured according to Jordan et al.[8], that is between magnesium atoms of chlorine rings, the center of the carbonyl ring of the phylloquinones, and centers of the iron–sulfur clusters, were at variance with that of the trimeric crystal structure (see Supplementary Table 3).

**Structure of PsaL**. In our single-particle cryo-EM structure of the biochemically induced PSI monomer, PsaL could be modeled with confidence to a completeness that is almost matching that of the trimer crystal structure (87% vs >95% of modeled residues). The main difference is that the C-terminal short α-helix involved in the formation of oligomer contacts is missing (Fig. 5a). Though it visualizes all other α-helices in PsaL, this short C-terminal α-helix is also not visualized in our X-ray crystal structure. Against our expectation and despite its peripheral location and prominent role in oligomerization, however, not only almost all residues of PsaL but also all three PsaL-bound chlorophylls could be modeled into this well-defined region of the density map. This demonstrates that monomerization does neither result in the overall flexibility of PsaL nor the loss of PsaL-bound chlorophylls. However, in support of its suggested structural role of the trimerization domain, the PsaL-bound carotenoid BC-L22 could not be found in our cryo-EM structure of the *T. elongatus* PSI monomer.

**Carotenoid network**. Carotenoids make up a major and important part of PSI organic cofactors. While a total number of 22 carotenoids was identified in the trimer crystal structure, we

modeled 26 carotenoids in our monomer structure. Among these, 8 were newly found and 4 of previously identified carotenoids were lost. Our spectroscopy results indicated a decrease in carotenoids in the monomer versus the trimer of the same preparation, which contrasts the larger number of carotenoids modeled in our monomer structure. This finding might indicate the presence of unmodeled carotenoids in the trimer crystal structure.

We superposed our monomer structure with the trimer crystal structure (PDB ID 1JB0) to localize new and lost carotenoids (Fig. 6). We observed that 18 of the 26 carotenoids in the monomer model coincided with their counterpart in the trimer structure. Lost carotenoids were BC-F4016 (a 13′-cis isomer) and BC-J4012 from the periphery of the trimer, BC-B4009 in the vicinity of the periphery and BC-L4022 from the trimerization interface. The majority of the newly found carotenoids (BC-A853, BC-A854, BC-A855, BC-A856, BC-A857, and BC-A858) were found in the perimeter of PsaA, and only one new carotenoid (BC-B844) was found in PsaB (Fig. 6). BC-A853 had also been modeled in the very recently reported single particle cryo-EM structure of the *T. elongatus* PSI trimer[20]. Among the newly modeled carotenes, one half can be assigned to the lumenal side and another half to the stromal side. All newly found carotenoids are located close to chlorophyll chlorine rings within a distance range of 3.5–5.1 Å. Among the new carotenes, only BC-A852 was fully modeled as a 13′-cis isomer. Likewise, BC-A856 was modeled as a 9′-cis isomer. Interestingly, out of these two carotenoids, the head group of BC-A852 is located in the same region occupied by the lost chlorophyll aC-K1401 of the trimer crystal structure and BC-A856 is in proximity to the same lost chlorophyll. Similarly, the new carotenoid BC-A855 is located in the middle of two lost chlorophylls, aC-A1402 and aC-K1401 (Supplementary Fig. 5). It is also notable that among the lost

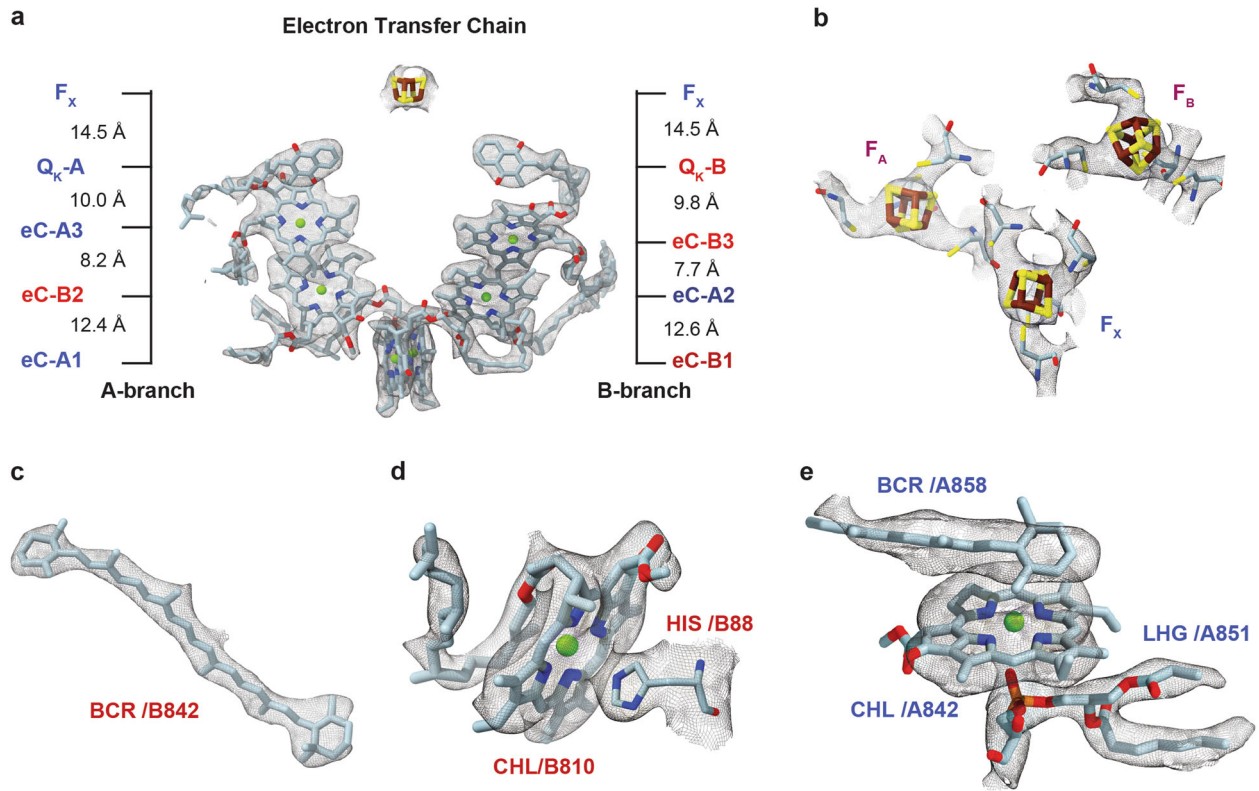

**Fig. 4 Selected cofactors and their corresponding 3D cryo-EM Coulomb density map. a** The electron transfer chain (ETC) of monomeric PSI viewed parallel along the membrane plane. Distances between the A-branch and B-branch measured between magnesium atoms of neighboring chlorophylls, magnesium and center of the carbonyl ring of neighboring chlorophylls and phylloquinones and the center of the $F_x$ iron–sulfur cluster and center of the carbonyl ring of the neighboring phylloquinones are indicated. **b** The three iron–sulfur clusters $F_x$, $F_A$, and $F_B$ of the PSI monomer. **c** A fully modeled carotene with visible polyene chain methyl bumps in the density map. **d** A chlorophyll coordinated by a histidine residue and its own phytol chain. **e** A chlorophyll coordinated by a lipid and a newly found carotenoid.

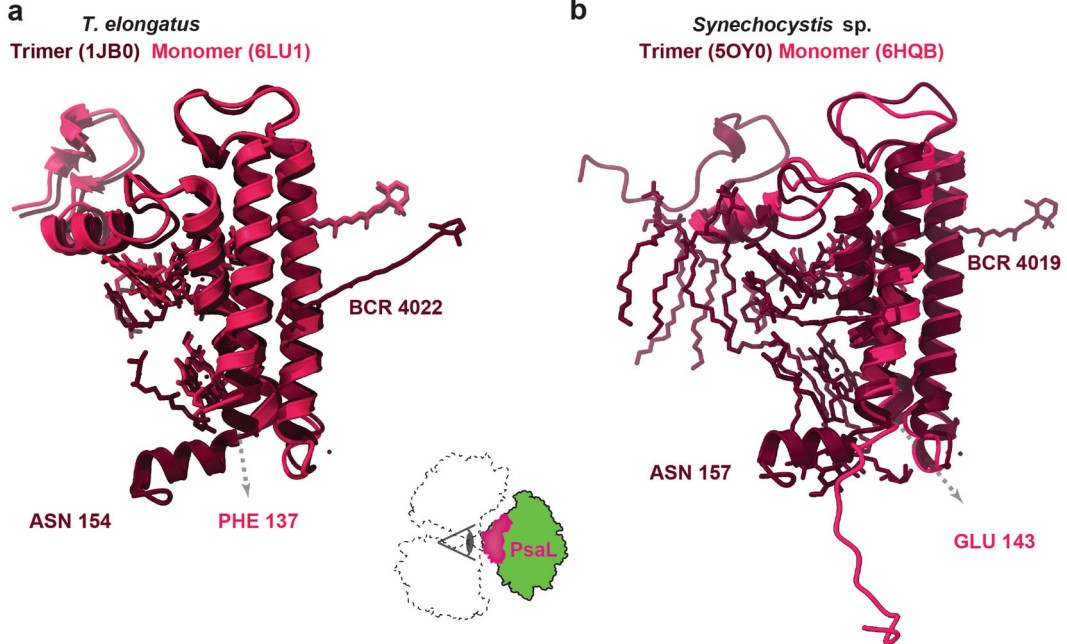

**Fig. 5 PsaL subunit as viewed from the center of the PSI trimer. a** *T. elongatus* trimer and monomer superposed for illustration of the absence of the C-terminal α-helix and the PsaL-bound carotenoid BCR4022. The C-terminal phenylalanine in our monomer model of PsaL is indicated by a dashed arrow. **b** *Synechocystis* sp. trimer and monomer superposed with the C-terminal glutamate of the monomer model of PsaL indicated by a dashed arrow.

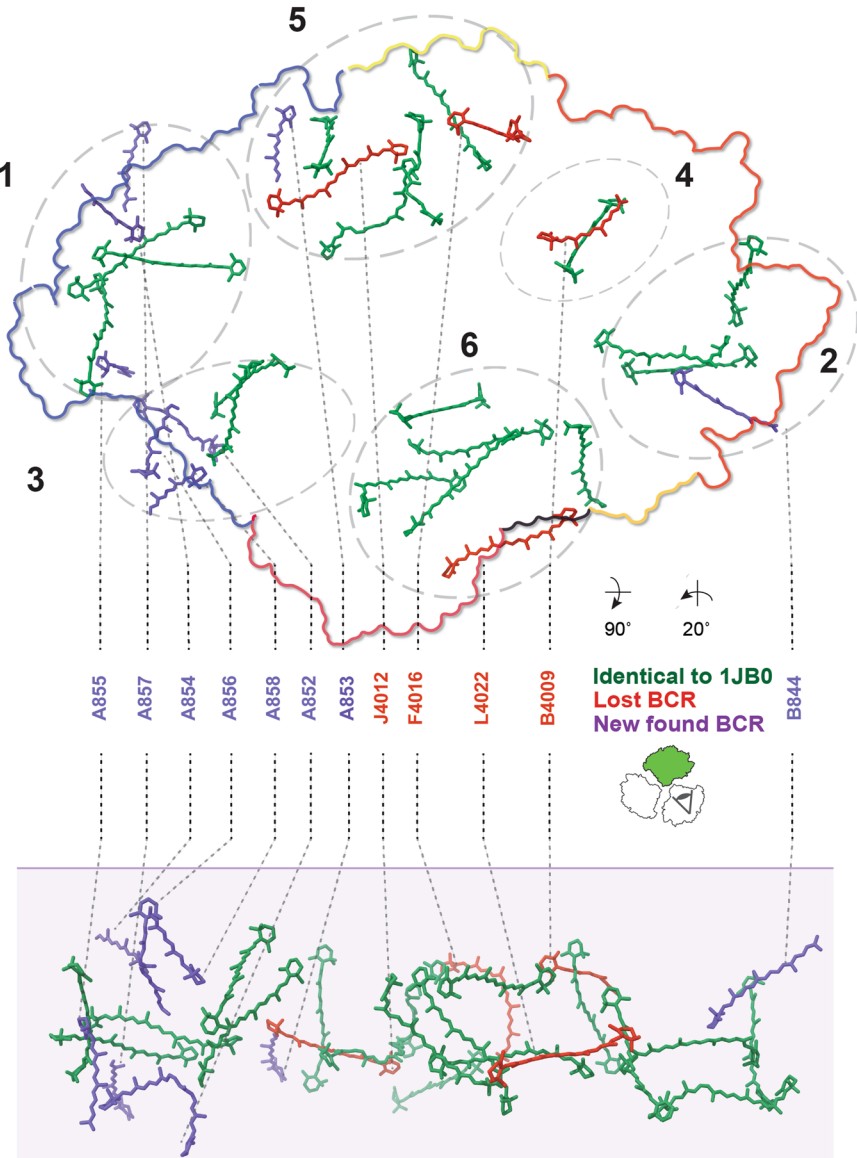

**Fig. 6 Carotenoid network.** View on the membrane normal from the stromal side (upper panel) with monomer outlined according to subunit color. Lower panel: side-view along the membrane plane with the membrane indicated in shade. Carotenoids found in both monomer and trimer structures are colored in green, carotenoids not found in the monomer structure are colored in red and newly found ones are colored in blue. Carotenoids are clustered according to Jordan et al.[8] and indicated by dashed circles and their respective number.

pigments, BC-L4022/aC-M1601, BC-F4016/aC-X1701, and BC-B4009/aC-B1220 were found in close range. Jordan et al.[8] grouped the carotenoids into six distinct clusters. The most noticeable change in our model regarding these clusters is that the balance of the number of carotenoids between the clusters 3/4 and the clusters 1/2 is now broken (Fig. 6).

**Chlorophyll network**. The number of red chlorophylls per PSI monomer has been estimated to be ~9 on the basis of spectroscopy, with ~4 of them being C708 chlorophylls and the remaining 4–5 of them being C719 chlorophylls, from their absorption peaks at 708 nm and 719 nm, respectively[40,46]. It is well known that two C719 chlorophylls are lost upon monomerization of trimeric *T. elongatus* PSI[47,48]. To identify which of the 96 chlorophylls of the PSI monomer were lost during the preparation, we superposed our cryo-EM model (PDB ID 6LU1) with that of the trimer crystal structure (PDB ID 1JB0; Fig. 7). We observed that 82 chlorophylls reticulating within the PSI

monomer fully coincided with their counterpart in the trimer crystal structure. However, 14 chlorophylls of the peripheral antenna present in the trimer crystal structure were absent in our model. Amongst these aC-A1402, aC-K1401, aC-J1302, aC-J1303, aC-F1301, aC-X1701, and aC-B1233 are located on the lumenal side of the peripheral antenna network and the remaining chlorophylls are all located on the stromal side of the peripheral network.

The majority of the 14 lost chlorophylls were from PsaB (7), whereas the remaining 7 were from PsaJ (2), PsaA (1), PsaF (1), PsaK (1), PsaM (1), and PsaX (1). Remarkably, only the lost chlorophylls of subunit PsaM (aC-M1601) and PsaK (aC-K1401) were situated at the monomer-monomer interface of the PSI trimer. Furthermore, against all expectations, the PsaL trimerization domain chlorophylls remained in our PSI monomer structure and their position was preserved. All other 12 lost chlorophylls were at the exterior membrane-facing side of the PSI trimer and thus accessible by the surrounding thylakoid lipid

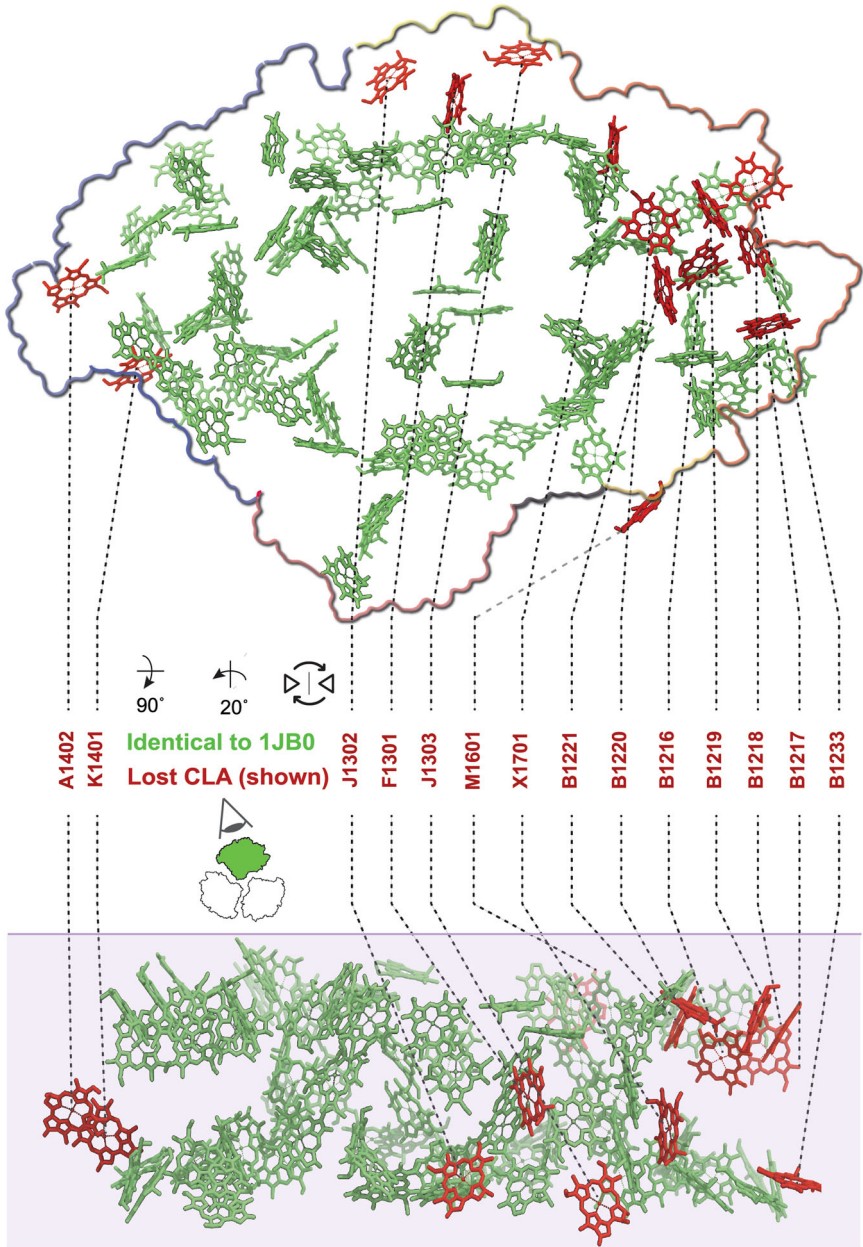

**Fig. 7 Chlorophyll network.** View on the membrane normal from the stromal side (upper panel) with monomer outlined according to subunit color. Lower panel: side-view along the membrane plane with the membrane indicated in shade. Chlorophylls found in both monomer and trimer structures are colored in green, chlorophylls not found in the monomer structure are colored in red.

bilayer. Among the 14 lost chlorophylls, only a group of six chlorophylls on the stromal side of PsaB, aC-B1216-B1221, were in contact with each other and formed a tight cluster (Fig. 8). The location of this chlorophyll cluster coincided with the unique region of the monomer PsaA/PsaB core that could not be modeled in our cryo-EM structure, the stromal side loop-helix-loop region PsaB 291–316 (Fig. 8 inset 2). In the trimer crystal structure, two phenylalanine residues (B309 and B310) from the short horizontal α-helix extend from above to form a loose clamp that holds on to the chlorine ring coordinating phytol chain of the membrane-facing chlorophyll B1219 (Fig. 8 inset 1). This hexameric cluster of chlorophylls was located above the characteristic chlorophyll trimer of PsaB that resembles a staircase. The outer 'step' of the staircase chlorophylls, aC-B1233, was also absent from our model (Fig. 8 inset 3). Interestingly, three of the chlorine rings in the hexameric cluster

and two of the 'staircase trimer' were coordinated by water. In the trimer crystal structure lipid IV, the only lost lipid in the monomer, is in contact with the hexameric cluster, sandwiched between the hexameric cluster and PsaX which is coordinating the headgroup of the phosphatidylglycerol with the side chain of Tyr-8.

## Discussion

In this work, we determined and analyzed the structure of PSI from the thermophilic cyanobacterium *Thermosynechococcus elongatus*, monomerized by LDAO and ammonium sulfate treatment. We succeeded in extracting PSI monomers at high yield. Characterization by biochemical methods, mass spectrometry, X-ray crystallography, and EM showed it to be active and monodisperse and to harbor all the expected subunits (Fig. 1 and Supplementary Figs. 2, 8, 9, and 10). However, the possibility of a

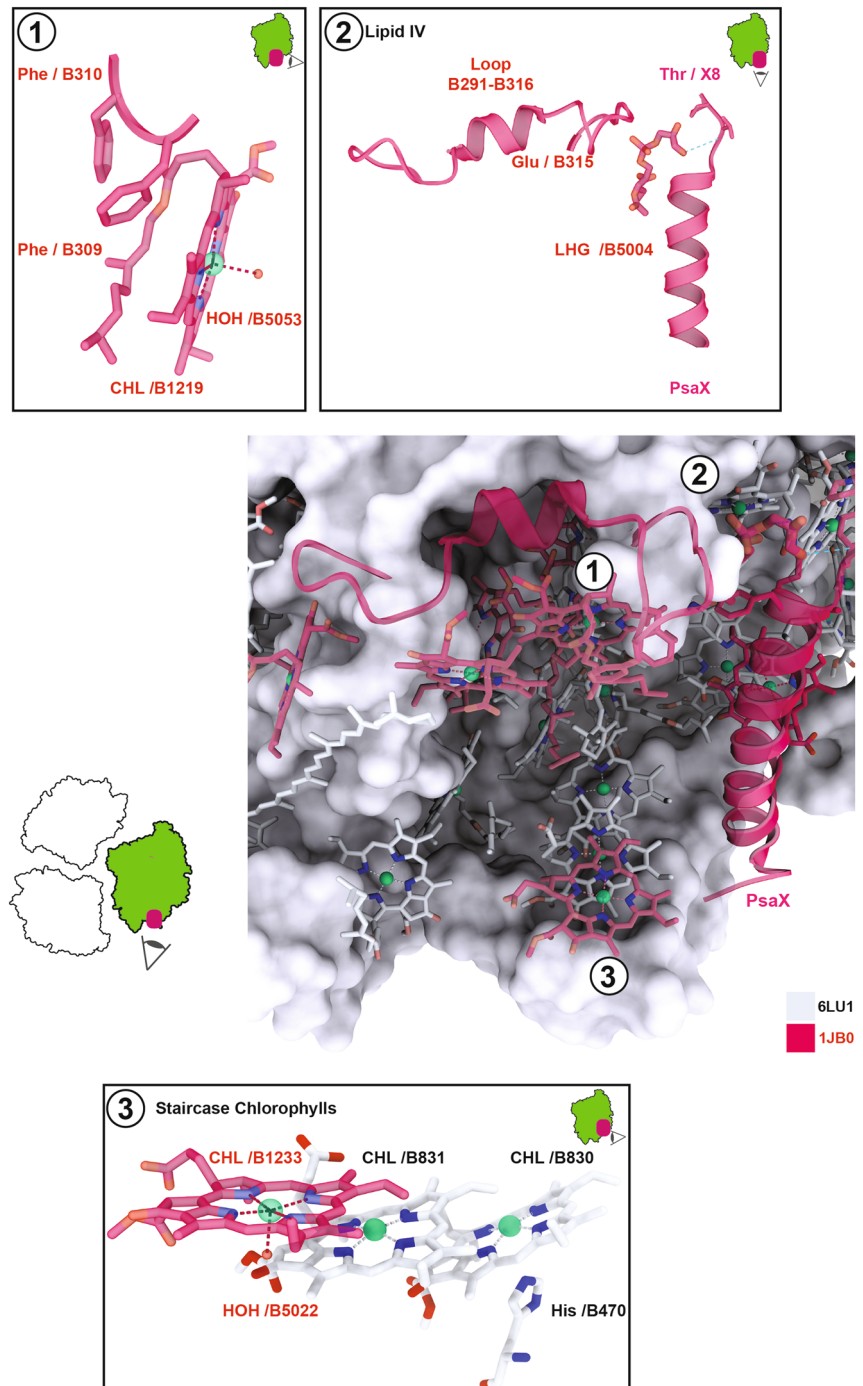

**Fig. 8 The main region of disorder in monomerized PSI and the putative site of lost red chlorophylls.** The monomer model 6LU1 is depicted in gray as surface for protein and as ball and stick for ligands. Ligands and protein regions of the trimer structure model 1JB0 that could not be modeled in our monomer structure in red ribbon or ball and stick. ① Weak coordination of the lost chlorophyll B1219 of the trimer structure. ② Interaction between the lost lipid IV of the trimer structure and the loop-helix-loop motif B291–B316 of PsaB (disordered in the monomer structure) and the lost PsaX subunit. ③ Weak coordination of the lost chlorophyll B1233 of the lumenal side trimer staircase chlorophylls.

partial removal of subunits or cofactors as a result of the LDAO treatment cannot be completely ruled out. In addition, we also observed the expected diminishment in absorption at 720 nm and a blue shift in the fluorescence spectrum of monomeric PSI relative to that of trimeric PSI. This provided us with a firm basis for structural analysis of monomeric PSI ensuring that structural features found are that of a bona fide PSI monomer. The 6.5-Å X-ray crystal structure of the PSI monomer within the tightly packed crystal state allowed us to visualize all the subunits previously found in the high-resolution trimer crystal structure at the α-helical level. This further corroborated that the monomerization process of our preparation method keeps the monomeric complex intact and does not lead to the loss of subunits. With this in mind, we carefully examined our high-resolution single particle cryo-EM monomer structure for changes compared to that of trimeric PSI. We found that PsaK and PsaX could not be modeled at all and PsaF and PsaJ only to about a fifth of the trimer crystal structure. These four subunits face the

membrane periphery of the PSI trimer and experience the longest exposure to the chemical and mechanical treatment during membrane solubilization, monomer isolation and cryo-grid preparation. Recently, the long-suspected notion that in single-particle cryo-EM experiments the vast majority of protein complexes adheres to the air–water interface, which can result in preferred orientations and harm the protein complex in question, has been conclusively demonstrated[49–51]. This specific limitation of single particle cryo-EM-based determination of protein complex structures potentially also applies to the regions of disorder observed for the *T. elongatus* PSI monomer. However, the same studies also showed that these air-water interface effects are highly protein-dependent and cannot be generalized. Therefore, for the present study it is highly informative that in the now reported single particle cryo-EM structure of the *T. elongatus* PSI trimer[20] (PDB ID 6TRA, EMDB ID EMD-10557) all subunits and co-factors that were found in the original X-ray crystal structure (PDB ID 1JB0) could be modeled. Moreover, the 6TRA model of trimeric *T. elongatus* PSI also included an improved model of PsaK. Importantly, it also visualized all chlorophylls and the protein regions of the here identified PsaX region of disorder in the monomer. Therefore, we believe that PsaF, PsaJ, PsaK, and PsaX are not physically lost during the purification or cryo-grid preparation and that these regions of the multisubunit complex are disordered in its free form in solution, as suggested by the relatively low resolution X-ray diffraction of our 3D crystals. Previously, cyanobacterial PSI had been monomerized and isolated using a wide range of approaches including genetic means like ΔpsaL mutants[52–54] or the addition of a His-tag to the C-terminus of PsaL[14] or treatment of solubilized PSI trimers by osmotic shock[36,55] and salt treatment at elevated temperature[41]. Interestingly, irrespective of the method used, monomerized PSI always exhibited the loss of red chlorophylls[30,47]. Why monomerization increases the susceptibility of the membrane-exposed surface region of *T. elongatus* PSI to lose red chlorophylls in comparison to the trimeric form is not clear. However, the reported reversibility of the process[56,57] points to an oligomeric state-dependent conformational disorder that could function as a molecular switch for red chlorophyll activity and thus as a cyanobacterial response to changing light conditions. With red chlorophylls of PSI in algae and plants being located in external antenna light-harvesting complexes, oligomerization as a molecular mechanism for controlling their activity becomes redundant.

A notable exception to the surprising invariance in PsaL after monomerization is the absence of its short C-terminal α-helix. This short α-helix, which is oriented roughly parallel to the membrane plane on the lumenal side, is believed to make crucial contacts with neighboring PSI monomers in the process of trimerization[8,16] or in the formation of tetrameric PSI[22–24]. This short α-helix is also absent in our monomer X-ray crystal structure and the same short α-helix has been reported to lose its secondary structure in the His-tagged C-terminus of the PSI monomer from *Synechocystis* sp. PCC 6803[14] (Fig. 5b). Possibly, the C-terminus of PsaL is disordered per se and only transforms into a more rigid α-helical secondary structure upon making contact with other disordered PsaL C-termini or the additional chelation of a $Ca^{2+}$ ion. To our surprise, the other regions of PsaL were well defined with all three PsaL-bound chlorophylls and one of the two bound carotenoids in the same position as in the trimer crystal structure. This is unexpected, as PsaL is clearly involved in the trimerization of PSI. Also, a breakup into monomers achieved by ΔpsaL mutants of PSI leads to the loss of PsaL and other chlorophylls[58]. The lack of instability and the absence of chlorophyll loss in the PsaL region of the PSI monomer refutes ideas on the location of C719 red chlorophylls in the

trimerization domain[8,30,59–61]. Since excitation transfer between monomers of the PSI trimer is thought to be minor[62], the lost carotenoid (L-4022) of PsaL probably occupies a structural role in oligomerization rather than a role in the excitation energy transfer between neighboring PSI monomers[11].

Among the carotenoids that were identified as new in our monomer structure, a number of them suggested mobility. For instance, the newly found carotenoid A855 was in a location that is occupied by PsaK in the trimer crystal structure and the newly found carotenoid A852 was partially in the same location as the PsaK-bound chlorophyll of the trimer crystal structure. Moreover, in our monomer structure conserved and retained carotenoid A844 shifted towards PsaK. Therefore, PSI-bound carotenoids appear to experience a relatively large degree of mobility that is possibly induced by the monomerization process. This apparent higher mobility of carotenoids is reflected by their relatively low Q-score[63] of 0.61 for all carotenoids and 0.53 for all newly found carotenoids in our model (see Supplementary Table 2 for Q-scores of all subunits, chlorophylls and carotenoids). Mobility might also underlie the asymmetric distribution of carotenoids between the monomers in the trimeric crystal structure of *Synechocystis* sp. PSI[16]. As we assume PsaK to be only disordered and not physically lost, the apparent clash of carotenoids with PsaK is likely avoided by an at least partial shift of this subunit upon monomerization. In support of a physiological role for this carotenoid, the newly found carotenoid BC-A853 was also observed in the cryo-EM structure of the *T. elongatus* PSI trimer[20]. The UV-Vis difference spectrum (Fig. 2b) suggests that diminished absorbance in the carotenoid region for the monomer sample is the consequence of loss of PSI-bound carotenoids. Therefore, we posit that carotenoids at the monomer–monomer interface of PSI, which are shielded from solubilization in the trimer, are lost during the purification process.

The biggest difference between our single particle cryo-EM structure of the PSI monomer and the trimer crystal structure lies with the chlorophylls. Of the 96 chlorophylls present in the trimer crystal structure, 14 were not found in our monomer structure and thus were either physically lost or are too disordered to permit a well-defined density map. It is well known that red C719 chlorophylls absorption is lost from the spectral properties of *T. elongatus* PSI trimers upon monomerization with an estimate of two C719 chlorophylls lost per monomer[30,48,64]. This typical loss of red chlorophylls was confirmed in our monomer preparation by the observed decrease in absorption around 720 nm and the blue shift of fluorescence emission. Additional support was provided by the precisely defined positions and numbers of lost chlorophylls in our monomer structure. This puts us now in a good position to discuss the possible locations of the C719 red chlorophylls of the *T. elongatus* PSI trimer. Previous proposals for the molecular identity of red chlorophylls were made on the basis of their relative distance to each other, alignment of $Q_Y$ to the membrane plane, the possible excitonic coupling of their chlorine rings, and site energies[8,35,37,59–61,65–71]. The observation that ΔpsaL mutants lead to both monomerization and the loss of red chlorophylls suggested the presence of C719 chlorophylls in the vicinity of the trimerization domain such as B1237/B1238 and A1132/B1207[59–61,66,69,70]. Along a similar line of arguments, it was proposed that since disruption of the trimer causes their loss, red chlorophylls should be located at the monomer–monomer interface[8]. Based on the idea that funneling of excitation energy to the reaction center is aided by close localization of red chlorophylls and that quenching of P700[+] necessitates a limited distance, vicinity of red chlorophylls to P700 had been suggested[72,73]. In a spectroscopic study of *T. elongatus* PSI, the detected remaining quantum yield of P700 oxidation at 5 K of

~50% was interpreted in favor of a clustered localization of red chlorophylls at the peripheral antenna[30]. The lack of chlorophyll loss in the trimerization region, especially in PsaL itself and at the monomer-monomer interface in our structure is a surprise and disfavors ideas on localization of red chlorophylls in these regions (Fig. 5). Further, the conservation of all core chlorophylls conflicts with the notion of red chlorophylls in the vicinity of the ETC. In agreement with a chemically induced loss, we find that with the exception of M1601 and K1401, all lost chlorophylls lie on the membrane-facing peripheral antenna region of the PSI trimer. In this distal region of PSI, the most important loss is that of seven PsaB chlorophylls in the vicinity of PsaX. In the PSI trimer, six of these form a tight cluster in contact with lipid IV bridging PsaX and the cluster (Fig. 8). The seventh lost chlorophyll B1233 is the outermost, membrane-facing 'stair' of the 'staircase' chlorophyll trimer on the lumenal side of PsaX[8]. The stromal side cluster is directly exposed to the bulk lipid of the surrounding thylakoid membrane. At the edge of the stromal cluster, the lost chlorophyll B1219 is particularly weakly coordinated by only a water on the protein side and its own phytol chain on the membrane side. It appears to be held in place by two phenylalanine side chains (PsaB309/310) of the stromal side loop-helix-loop motif of PsaB (PsaB291–316), which is the only disordered region of the PsaA/PsaB heterodimer in our monomer structure. The peripheral, lipid-bilayer exposed location, and its weak mechanical coupling to PSI puts B1219 into an ideal position to receive thermal energy from surrounding lipids. This remarkably loose coordination and spring-like architecture makes B1219 an excellent candidate for a red chlorophyll. This is in line with the suggestion that the chlorophyll pair B1218/B1219 is part of the red chlorophylls based on spectroscopy studies of monomeric PSI from *T. elongatus*[35,69]. Likewise, the outermost staircase chlorophyll B1233 is also weakly coordinated by water on the protein side and is directly exposed to the membrane, ideally situated to receive thermal energy from bulk lipids.

Interestingly, the triple 'staircase' chlorophylls B1231, B1232 and B1233 have been previously proposed as red chlorophylls on the basis of their spatial arrangement[8,35,61,66,69,74] that is thought to be ideal for excitonic coupling. Also, as mentioned above, spectroscopy of *T. elongatus* PSI performed at 5K suggested that the C719 red chlorophylls are localized in a single cluster of the peripheral antenna. Our structural findings on the location of lost chlorophylls combined with the spectroscopic evidence that red chlorophylls should be present as a cluster[30,35] leads us to conclude that the PsaX-adjacent chlorophyll cluster is the molecular identity of red chlorophylls in *T. elongatus* PSI. Given that the trapping kinetics of cyanobacterial PSI are dominated by red chlorophylls[40,47], the clustering of the red chlorophylls at the periphery of PsaB distal to PsaA might provide a simple explanation for asymmetric ETC branch use in cyanobacteria versus symmetric branch use in PSI of algae and higher plants[75,76].

PSI from the mesophilic cyanobacterium *Synechocystis* sp. is the only cyanobacterial Photosystem I for which both trimer and monomer crystal structures have already been reported[14,16]. Notably, in contrast to *T. elongatus* PSI, *Synechocystis* sp. PSI does not contain C719 red chlorophylls. This prompted us to compare the region of lost chlorophylls adjacent to PsaX from *T. elongatus* with the same region of the PSI complex of *Synechocystis* sp. in both trimer and monomer structures (Supplementary Fig. 7). Important differences are the absence of PsaX and the absence of B1233 with its supporting loop region of B490–496 in *Synechocystis* PSI[16]. Interestingly, these differences had been noticed even before the arrival of the first *Synechocystis* crystal structure[64]. Moreover, the loop formed by the residues B306–316 of the loop-helix-loop region B291–316 in the *T. elongatus* PSI trimer structure, that with its two coordinating phenylalanine side

chains is stabilizing the very weakly bound B1219, disordered in our monomer structure, is also missing in *Synechocystis* sp. PSI. Instead, *Synechocystis* sp. PSI has an additional third chlorophyll which, if present in PSI of *T. elongatus*, would clash with PsaX.

Structural differences of *T. elongatus* PSI with *Synechocystis* sp. PSI are concentrated in the main region of disorder in monomerized *T. elongatus* PSI. And the most striking spectroscopic difference between the two species lies with the absence of C719 chlorophylls in *Synechocystis* sp. PSI. This provides strong support for the assignment of B1233 and B1219 as C719 red chlorophylls. For strong red shifts in chlorophylls excitonic coupling alone is thought to be insufficient and the importance of magnesium coordination by water has been suggested as a factor allowing stronger red shifts[64,77]. We find that the absence of the water-coordinated 'outer staircase' chlorophyll B1233 and the replacement of water coordination by histidine coordination in B1219 of *Synechocystis* PSI is in support of this notion (see also Supplementary Fig. 7). In a very recent study, a chimeric *Synechocystis* PCC 6803 PSI into which the lumenal PsaB loop supporting the outer staircase chlorophyll B1233 was introduced has been characterized by both cryo-EM and spectroscopy[21]. The cryo-EM based atomic model showed that an additional chlorophyll corresponding to B1233 is present in the chimeric PSI complex and is causing a red shift. These findings are also in strong support for our assignment of B1233 as a red chlorophyll and of the PsaX region as the site of the red chlorophyll cluster in *T. elongatus* PSI.

The position and architecture of the PsaX-adjacent chlorophyll cluster should make these chlorophylls excellent receivers of thermal energy from the vibrational bath of the surrounding lipid bilayer, allowing uphill energy transfer from these long-wavelength antenna pigments to P700. On this basis, we propose that the function of PsaX and lipid IV together with PsaB291–316 is to stabilize the spatial arrangement of these red chlorophylls while allowing the mechanical vibration necessary for efficient reception of thermal energy. To our knowledge, the molecular source of thermal energy for red chlorophylls has not yet been discussed in the literature. Since water molecules as the most common energizer of Brownian machines[78] are not available for the hydrophobic chlorophylls, and lipids are an established source of thermal energy for membrane-bound molecular machines[79], this is quite plausible. A possible scenario for the observed loss of these red chlorophylls upon monomerization is that lipid IV is extracted into the bulk solution in the first step. This then leads to the destabilization of both PsaX and the stromal side loop-helix-loop motif of PsaB 291–316 and culminates in the disorder of the seven chlorophylls. The region we identified as the main location of red pool chlorophylls has a cave-like structure that provides space for clustered pigments in direct contact with the embedding membrane. In this hollow structure B1233 on the lumenal side, the phytol chain of B1216 in the center and B1219 on the stromal side all protrude straight into the bulk lipid phase in an antenna-like fashion. Forming the contact of the cluster with the membrane lipids, these three pigments comprise three different conformations that could be interpreted as three unique modes of interaction with the membrane. Perhaps these three pigments in their common exposure to the rapid movement of lipid acyl chains but with divergent microarchitecture are ideally situated to receive the different modes of vibrational thermal energy provided by the lipid bath of the surrounding membrane.

In summary, comparison of the trimer crystal structure (PDB ID 1JB0) with our monomer crystal structure (PDB ID 7WB2) and single-particle cryo-EM monomer structure (PDB ID 6LU1) allowed us to examine the candidates of red chlorophylls and to conclusively assign them in the structure. Based on our findings

of the assignment of red chlorophylls and the position-specific disorder and flexibility of the oligomeric integral membrane protein complex, we propose a functional and structural role of PsaX in the stabilization of red chlorophylls. For the determination of PSI monomer structures at truly atomic resolution, it is likely more meaningful to make full use of recent advances in high-resolution single-particle cryo-EM techniques that employ electron cryomicroscopes (cryoTEMs) equipped with an electron source of smaller energy spread than that of conventional cryo-TEMs[80–82].

## Methods

**Thylakoid membrane preparation.** 20 L of a *Thermosynechococcus elongatus* BP-1 wild-type culture (grown in *Airlift* fermenter) was concentrated to a volume of 2 L using an Amicon® ultrafiltration cell (Amicon® DC 10 LA). The cells were sedimented (10 min, 4 °C, 9,000 g, JLA 8.1000 rotor) and washed with 150 ml of buffer A (20 mM MES pH 6.5, 10 mM MgCl$_2$, 10 mM CaCl$_2$). Subsequently, the sediment was resuspended in further 200 ml of buffer A with 0.2% (w/v) lysozyme and incubated with agitation at 37 °C for 90 min in darkness. All the following steps of the preparation of thylakoid membranes were performed under darkness to avoid generation of reactive oxygen species by the photosystems. The cells were incubated in a pre-cooled parr-bomb system at 2000 psi and 4 °C for 20 min and disrupted during the subsequent pressure drop. To remove soluble cell components, the cell debris and membranes were sedimented (10 min, 4 °C, 15,000×g, JLA 16.250), washed three times with buffer A and subsequently with buffer B (20 mM MES pH 6.5, 10 mM MgCl$_2$, 10 mM CaCl$_2$, 500 mM mannitol). At this point, the sedimented membranes were resuspended in 100 ml of buffer B with 20% (v/v) glycerol, frozen in liquid nitrogen, and stored at −80 °C.

**PSI monomer isolation.** Thylakoid membranes of *T. elongatus* were washed with buffer B (30 min, 4 °C, 20,000 × g) and subsequently with buffer B supplemented with 0.05% (w/v) (β-DDM). After sedimentation, the supernatant containing phycobilisomes to a large extent was discarded and the pellet was resuspended in extraction buffer (20 mM HEPES pH 7.5, 10 mM MgCl$_2$, 10 mM CaCl$_2$, 200 mM ammonium sulfate (AMS)) at a final Chl concentration of 1 mg Chl/ml, supplemented with 0.5% (w/v) LDAO and incubated with gentle agitation at 20 °C for 30 min. After incubation, the samples were sedimented (60 min, 4 °C, 180,000×g, Type 70 Ti) and the PSI-containing supernatant was applied onto a discontinuous sucrose gradient (45 ml of 14% (w/v) sucrose in buffer B + 0.03% (w/v) β-DDM, underlaid with a 5-ml 80% (w/v) sucrose cushion in buffer B + 0.03% (w/v) β-DDM, in Type 45 Ti centrifugation tubes) and fractionated by ultracentrifugation (18 h, 4 °C, 123,000×g, Type 45 Ti, with slow acceleration and deceleration). Chlorophyll containing sections of the sucrose gradient were collected, mixed with 2 M AMS and applied on a hydrophobic interaction chromatography (HIC) column (POROS 50 OH, CV 53 ml, Thermo Fisher Scientific, USA) integrated into a high-performance liquid chromatography (HPLC) system (ÄKTA purifier, GE Healthcare, Sweden). The elution of PSI monomers was achieved by a decreasing AMS gradient of 1.65 M to 0 M AMS over 6 column volumes. The eluted fractions were concentrated to a volume of 10 ml using a stirred cell (AMICON 8050, MWCO: 100 kDa, Ultracel-100 membrane, EMD Millipore Coop., USA) and, to remove excess AMS, applied on a desalting column (HiPrep 26/10 Desalting, GE Healthcare, Sweden). Purified sample peaks were then used for activity measurements and analytical gel electrophoresis.

**PSI activity measurement.** PSI activity was indirectly measured using the Mehler reaction as previously described[41] via the time-dependent oxygen consumption, detected optically with an oxygen sensor system (PST3, FIBOX 3 (PreSens™)). Monomeric or trimeric PSI was added to 1 ml of activity buffer (25 mM HEPES pH 7.5, 500 mM NaCl, 3 mM MgCl$_2$, 330 mM Mannitol) at a final concentration of 5 µg Chl/ml, supplied with 0.8 mM dichloro-phenol-indophenol (DCPIP) and 5 mM sodium ascorbate as sacrificial electron donors and 1.2 mM methyl viologen as electron acceptor. The sample mixture was prepared inside a tempered, opaque cuvette at 30 °C under constant stirring. After an initial incubation of 2 min in the dark the measurement was started by switching on the measuring light ($\lambda = 680$ nm) for 3 min. The PSI-dependent oxygen consumption was calculated via the linear slope and specified as:

$$\mu\text{mol O}_2 \cdot \text{mg Chl}^{-1} \cdot \text{ml}^{-1} \cdot \text{h}^{-1}$$

**Blue native polyacrylamide gel electrophoresis.** Different oligomeric forms of PSI as well as their purity were determined via blue native polyacrylamide gel electrophoresis (BN-PAGE, modified as described previously[83]). The separating gel consisted of an acrylamide gradient (3.5% to 16% of 30% (v/v) acrylamide/bis-acrylamide 32:1) and was polymerized by 10% (v/v) APS with TEMED. After the addition of BN-sample buffer (5% (v/v) glycerol, 0.01% (w/v) Ponceau S) to the samples, they were applied on the gel and separated in an electric field at an initial voltage of 100 V. The inner gel chamber was initially filled with blue cathode buffer

(50 mM Tricine, 15 mM Bis-Tris, 0.02% (w/v) Coomassie BB G-250), while the outer chamber contained anode buffer (50 mM Bis-Tris). After 30 min, the blue cathode buffer was exchanged with clear cathode buffer (50 mM Tricine, 15 mM Bis-Tris) to discolor the gel. The voltage was raised to 150 V for about one hour until an adequate separation of the protein bands was achieved.

**SDS-PAGE and two-dimensional polyacrylamide gel electrophoresis.** To analyze the subunit composition of the isolated proteins, the standard method of polyacrylamide gel electrophoresis was carried out as described previously[84]. The samples were mixed with SJ-sample buffer, heated at 60 °C for 30 min, and, after a short centrifugation, applied on the polymerized SDS gel. Moreover, the SJ-PAGE was used as a second dimension to further analyze the natively separated complexes via BN-PAGE. Therefore, each BN gel strip was cut from the gel and incubated in denaturing buffer (125 mM Tris pH 6.8, 73% (w/v) urea, 64% (w/v) glycerol, 9% (w/v) SDS, 0.2% (v/v) β-mercaptoethanol) for 1.5 h at room temperature and with gentle agitation. After incubation, the stripes were washed in SJ cathode buffer (0.1 M Tris, pH 8.9, 0.1 M Tricine, 0.1% (w/v) SDS) and carefully installed on top of a prepared 2D-PAGE separating gel (one strip per gel). The stripes were then coated with SJ stacking gel and a protein standard (PageRuler Prestained, Thermo Scientific™) was applied on every gel. The separation of the subunits was achieved by an electric field at 35 mA per gel in the PROTEAN® II xi gel chamber (BioRad Laboratories, Inc.) in SJ anode buffer (0.1 M Tris pH 8.9, in outer gel chamber) and SJ cathode buffer (inner gel chamber).

**UV/Vis spectroscopy.** The absorption spectra were measured in almost complete absence of free detergent for GraDeR[43]-prepared PSI monomer and trimer using a UV-2450 spectrophotometer (SHIMADZU) in the range between 300 nm and 750 nm (Supplementary Data 1).

**Fluorescence emission spectra at 77 K.** Fluorescence emission spectra were measured with a JASCO FP-6500 spectrofluorometer. Trimeric and monomeric PSI complexes were diluted in buffer B with 0.03% (w/v) β-DDM to a final concentration of 120 µg/ml. Subsequently, the diluted samples were flash-frozen in liquid nitrogen and immediately used for the measurements. P700 excitation was achieved by an actinic light at 436 nm while fluorescence emission spectra were measured from 650 to 780 nm with a step size of 1 nm and a bandpass filter of 5 nm (Supplementary Data 2).

**Matrix-assisted laser desorption/ionization time-of-flight mass-spectrometry (MALDI-TOF MS).** The mass spectrometric analysis of PSI monomer and trimer was carried out with the Ultraflex III, MALDI-TOF/TOF-mass spectrometer (Bruker). Initially, the samples were treated with a saturated matrix solution of sinapic acid (sinapic acid in 0.1% (w/v) trifluoroacetic acid and 34% (v/v) acetonitrile). Aliquots of 2 µl of a 1:1, 1:5, and 1:10 diluted purified sample with the matrix were applied on a MALDI-TOF target and air-dried for 30 min. The isolated proteins were then ionized by the internal MALDI 200 nitrogen-laser, creating a 337-nm light pulse with an output power of 150 kW. By applying a high voltage, the mass of the peptides could be detected by their time-of-flight in the high vacuum.

**Negative stain electron microscopy.** Negative stain EM was used to assess sample quality of PSI monomer preparations. An aliquot of 3.5 µl ~100x diluted protein solution was applied to glow-discharged (10 s at 5 mA) continuous carbon film-coated copper grids (Nisshin EM) and stained using a 2% (w/w) uranyl acetate solution. After brief blotting (Whatman #1) the grid was air-dried and examined on an H-7650 HITACHI electron microscope at 80 kV, equipped with a 1 × 1 k Tietz FastScan-F114 CCD camera (TVIPS, Gauting, Germany).

**Cryo-EM image data collection.** Detergent exchange from β-DDM to GDN and removal of excess free detergent was achieved using the GraDeR approach[43]. Briefly, PSI monomer fractions were layered onto a sucrose/GDN step gradient (20 mM HEPES-NaOH, pH 7.0, 10 mM MgCl$_2$) starting and ending with 0.1 M and 1.3 M sucrose and 0.02% and 0.003% (w/v) GDN respectively. Gradients were centrifuged at 97,000×g (average g-force) for 18 h at 4 °C using a P40ST rotor and a CP70MX ultra-centrifuge (Hitachi, Koki). PSI monomer fractions were freed of sucrose and concentrated to 5 mg/ml using Amicon Ultra-100k concentrators. Aliquots of 10 µl were flash-frozen in liquid N$_2$ and stored at −80 °C for later use. An aliquot of 2.6 µl PSI monomers at 5 mg/ml in a buffer of 10 mM MgCl$_2$, and 20 mM HEPES (pH 7.0) and 0.003% (w/v) GDN was applied to glow-discharged (JEC-3000 FC, 30 s) Quantifoil holey carbon-supported copper grids (R 1.2/1.3, 300 mesh), blotted (Whatman #1) and plunge-frozen in liquid ethane using an FEI Vitrobot Mark IV at 4 °C and 95% humidity. Image acquisition was performed on a CryoARM200 (JEOL) equipped with a field emission gun and operated at 200 kV using flood beam illumination in bright field imaging mode. Movies were recorded using a K2 Summit detector (Gatan) in counting mode at a nominal magnification of 60,000× at the camera level, corresponding to a pixel size of 0.89 Å with 60 frames at a dose of 1.34 e$^-$/Å$^2$ per frame and an exposure time of 12 s per movie

resulting in a total dose of 80.4 e$^-$/Å$^2$. A total of 1530 movies were collected in series within a defocus range of 0.5 μm to 3.5 μm using JADAS software (JEOL)[85].

**Cryo-EM image processing**. All image processing was performed using RELION 3.0[44], cryoSPARC[86], and UCSF CHIMERA[87] on a GPU workstation (2 GPUs). Movie frames were aligned and summed using MotionCor2[88]. Estimation of the contrast transfer function (CTF) was performed using CTFFIND4[89]. After careful examination of all micrographs based on the quality of the Thon rings and the presence of contamination or incorrect imaging position, ~30% of the micrographs were discarded. The remaining 1,107 good micrographs were used for further image processing. A total number of 1,217,859 particles were automatically picked based on the Laplacian of Gaussian and extracted using a box size of 256 pixels. Extracted particles were subjected to reference-free 2D classification and good 2D classes were used as templates to autopick 227,910 particles. Particles on the carbon edge were manually removed resulting in 182,018 particles for extraction. Extracted particles were subjected to several rounds of 2D classification to discard bad particles, particles from carbon regions, and 2D classes exhibiting strong features from neighboring particles resulting in good 2D classes containing a total of 46,515 particles. All particles from the good 2D classes were used for de novo initial model building in cryoSPARC. The initial model was imported into Relion and used as a reference for 3D classification which resulted in further elimination of 410 particles. The remaining 46,105 particles were refined against the initial model followed by CTF-refinement and Bayesian polishing. This was iteratively repeated until postprocessing showed no further improvement in applied B-factor and resolution as monitored using the Gold Standard FSC criteria (Supplementary Figure 3). These iterative per particle corrections improved the B factor from −122 to −45. Additional focused classification trials on low-density regions did not yield any improvements. Local resolution was calculated by ResMap[90] in RELION on the full density map without masking.

**Atomic model building and refinement**. Coot 0.8.9.1[91], Phenix 1.14–3260[92], and UCSF Chimera/ChimeraX[87,93] were used for fitting, modeling, refinement and visualization. First, the *T. elongatus* PSI X-ray crystallography-based atomic model (PDB ID 1JB0) was adopted as a starting model. Fitting of 1JB0 to the final cryo-EM Coulomb map was done in Chimera and manual corrections were performed in Coot for minimization of rotamer, geometry, and Ramachandran outliers. Next, the fitted 1JB0 model was refined in real space using CSD Mogul substructure restraint files in Phenix followed by another round of manual corrections in Coot. This was repeated until validation statistics consolidated. Next, all parts of the model where the density map was of insufficient quality were erased and another round of real space refinements and manual corrections was carried out. When the validation statistics did not improve any further, a final round of real space refinements and manual corrections was performed using more accurate ligand restraint files. For beta carotene and chlorophyll *a*, restraint files calculated by the Grade Server (http://grade.globalphasing.org)[45] and for the iron–sulfur cluster, recently deposited restraint files created by Moriarty and Adams (sourceforge.net/ projects/geostd/)[94] were employed. Resolvability and Q-scores were calculated using MapQ, a USCF Chimera plugin[63].

**Statistics and reproducibility**. No statistical method was used to predetermine the sample size, and the experiments were not randomized. The investigators were not blinded to allocation during experiments and outcome assessment. The purification of PSI monomers was repeated over twenty times, which showed same results. The spectroscopic analysis was performed at least two times, and similar results were obtained. The cryo-EM data was collected from one grid. Individual images with bad ice were excluded from the data set by visual inspection. Data collection, processing and refinement statistics are summarized in Table 2.

**Reporting summary**. Further information on research design is available in the Nature Research Reporting Summary linked to this article.

## Data availability

Raw cryo-EM images of the *T. elongatus* PSI monomer after motion correction were deposited in the Electron Microscopy Public Image Archive (EMPIAR), under the accession number EMPIAR-10352. The cryo-EM density map of the *T. elongatus* PSI monomer has been deposited in the Electron Microscopy Data Bank (EMDB), under the accession number EMD-0977. The atomic coordinates of the *T. elongatus* PSI monomer cryo-EM structure and X-ray crystal structure have been deposited in the Protein Data Bank (PDB), under the accession number 6LU1 and 7WB2 respectively. All relevant data is available from the authors upon request.

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

## Acknowledgements

We are grateful for additional support and valuable scientific input for this project by Yuko Misumi, Jiannan Li, Hisako Kubota-Kawai, Takeshi Kawabata, Mian Wu, Eiki Yamashita, Atsushi Nakagawa, Volker Hartmann, Melanie Völkel and Matthias Rögner. Parts of this research were funded by the German Research Council (DFG) within the framework of GRK 2341 (Microbial Substrate Conversion) to M.M.N., the Platform Project for Supporting Drug Discovery and Life Science Research [Basis for Supporting Innovative Drug Discovery and Life Science Research (BINDS)] from AMED under grant number JP20am0101117 (K.N.), JP16K07266 to Atsunori Oshima and C.G., a Grants-in-Aid for Scientific Research under grant number JP 25000013 (K.N.), 17H03647 (C.G.) and 16H06560 (G.K.) from MEXT-KAKENHI, the International Joint Research Promotion Program from Osaka University to M.M.N., C.G. and G.K., and the Cyclic Innovation for Clinical Empowerment (CiCLE) Grant Number JP17pc0101020 from AMED to K.N. and G.K.

## Author contributions

G.K. and M.M.N. initiated the study; G.K., M.M.N., and C.G. designed research; O.Ç., A.F., H.T., A.K., E.E., T.K. performed experiments; O.Ç., A.F., H.T., and C.G. analyzed the data; K.N. supervised research and C.G., O.Ç., A.F., M.M.N., and G.K. wrote the manuscript with input from all authors.

## Competing interests

The authors declare no competing interests.
