## [Peer Review File · Communications Biology]

Reviewers' comments:

Reviewer #1 (Remarks to the Author):

This is a manuscript that describes the structure and properties of the monomeric form of photosystem I from the cyanobacterium *T. elongatus*. The authors have devised an efficient method for purification of the monomeric form of PSI, which is itself a useful contribution. They focus on the differences in pigment content between the monomer and trimer, and find that a number of chlorophylls are lost in the monomer and some carotenoids are observed in the monomer that were not observed in the trimer. Presumably, the carotenoids were there in the trimer as well and were for unknown reasons not observed. I would have liked some discussion of this point, especially since the spectra in Fig. 2 indicate that some carotenoid is lost upon monomerization.

The major conclusion of the manuscript is that the loss of "red" chlorophylls that is well documented to happen with monomerization are not pigments that are adjacent to the trimeric interface, in particular those in proximity to the PsaL subunit that is involved in trimerization. This is an interesting and somewhat surprising finding.

I think the weakest part of the paper is the spectroscopy. The loss of red pigments is not well documented. A careful difference spectrum between the monomeric and trimeric forms should be presented. It is also not clear whether the "lost" chlorophylls are really lost or just disordered. It should be an easy job to measure the total number of pigments in the preparation by extraction of a sample and take the spectrum of the extracted pigment and then measure the P700 content by either light-induced or chemically induced spectroscopy. Using the known extinction coefficients of chlorophyll and P700 will give the number of chlorophylls per P700. Carotenoids can be similarly measured. I think this needs to be done to put the whole question of the number of lost pigments on a firm footing.

Reviewer #2 (Remarks to the Author):

The manuscript by Coruh et al. reports the structural analysis of the monomeric PSI complex from *T. elongatus*. Monomeric PSI is characterized by the loss of red chlorophylls, which are present in the trimer. Several candidates of the red chl clusters were suggested, including chlorophyll clusters at the monomer-monomer interface. However, no conclusive assignment could be made till now. In this work, the authors solved structures of monomeric PSI, using the cryo-EM method (3.2 Å) and x-ray crystallographic method (6.5 Å). Compared with the structure of trimeric PSI from the same strain, They found: 1) PsaK and PsaX could not be modeled in their cryo-EM structure, and PsaF and PsaJ were only built to about a fifth of the trimer crystal structure. However, the electron density of the transmembrane helices of these subunits were observed in the low-resolution crystal structure. 2) the monomer structure contains eight carotenoids located at novel sites, which are absent in the trimer, while four carotenoids in the trimer are absent in the monomer. 3) the monomer contains 14 less chlorophylls compared with the trimer, therefore some of these chlorophylls might be responsible for the red spectra of the PSI trimer. The authors suggested a cluster of chlorophylls adjacent to PsaX contributes to the red spectra of PSI trimer.

Overall, this is an interesting contribution. However, there are some points that I would like the authors to address.

- Previously, a structure of monomeric PSI from *Synechocystis* 6803 was reported, it is encouraged that the authors add several sentences describing the comparison of the two monomer structures.
- In order to obtain the monomeric PSI, the authors solubilized thylakoid membrane with LDAO, and purified the PSI monomer through HIC. The harsh treatment may result in the loss of the peripheral subunits, such as PsaF, PsaJ, PsaK, and PsaX. This possibility should be stated in the manuscript.
- Line 390-392, It is obvious that PsaF, PsaJ, PsaK, and PsaX are not physically lost during the

purification, however, it is possible that they are dissociated from other core subunits during the grid preparation, which happens frequently in the cryo-EM sample preparations.

- Based on the text (Line 310-315, 429-432) and Supplementary Fig. 5, the newly found carotenoids (852, 855 and 856) seem to occupy the site of PsaK. However, the authors clearly explained that PsaK is present in the complex. Therefore it is difficult to imagine how these carotenoids bind at the PsaK site.
- The spectroscopic results indicated a decrease of carotenoids in the monomer versus the trimer, while the structural analysis showed larger number of carotenoids modeled in the monomer. Is there any explanation for the contradictory between the spectroscopic and structural results?

Minor points:

- The authors identified eight more carotenoids in the monomeric structure compared with the trimeric structure. It's better to show the cryo-EM densities of these carotenoids.
- Please provide the details PSI activity (Line 174-176) measurement in the METHODS.
- Line 315, Supplementary Figure 4 should be Supplementary Figure 5.
- Line 119, ref. 22 reports the tetrameric PSI from *Chroococidiopsis* sp TS-821, not from *Anabaena* 7120.

Reviewer #1 (Remarks to the Author):

This is a manuscript that describes the structure and properties of the monomeric form of photosystem I from the cyanobacterium *T. elongatus*. The authors have devised an efficient method for purification of the monomeric form of PSI, which is itself a useful contribution.

Thank you very much for stressing on the efficiency of our monomer preparation method. We believe our new purification will be of use for many groups involved in the study of monomeric cyanobacterial photosystem I.

They focus on the differences in pigment content between the monomer and trimer, and find that a number of chlorophylls are lost in the monomer and some carotenoids are observed in the monomer that were not observed in the trimer. Presumably, the carotenoids were there in the trimer as well and were for unknown reasons not observed. I would have liked some discussion of this point, especially since the spectra in Fig. 2 indicate that some carotenoid is lost upon monomerization.

- In our single particle cryo-EM structure of *T. elongatus* photosystem I at 3.2 Å overall resolution, we were able to model 8 carotenoids that were not found in the 2.5 Å X-ray crystal structure (PDB ID: 1jb0) of trimeric *T. elongatus* photosystem I by Jordan et al. published in 2001. As reviewer #1 rightly points out, this raises the interesting question, if the newly found carotenoids are truly new in the sense that they are specific to our monomer preparation. Or if perhaps they were also present in the PSI trimer of the crystals that were used for X-ray diffraction leading to the 2.5 Å resolution X-ray structure and simply could not be modelled. Although we cannot be 100% certain, we believe that both is true. While our manuscript was being reviewed, the first single particle cryo-EM structure of trimeric PSI from *T. elongatus* was published by the 'Berlin Group' Kölsch *et al.* at 2.85 Å overall resolution (Kölsch, A., *et al.* "Current limits of structural biology: the transient interaction between cytochrome c_6 and photosystem I." *Current Research in Structural Biology* (2020); PDB ID: 6tra; EMD ID: EMD-10557). When comparing their model (6tra) with ours (6lu1), we found that among our newly built 8 carotenoids, seven do not have a match in 6tra while one carotenoid (BCR A853 our notation, see Fig. 6) is matched by the newly found carotenoid BCR A851 (notation used in 6tra). All 8 newly modeled carotenoids in our monomer structure can be found at the periphery of the complex and in coordination of chlorine rings. Carotenoids are highly hydrophobic and known for their affinity to chlorine rings. During solubilization of the thylakoid membranes, many carotenoids may be co-solubilized and thereafter be present in free detergent micelles throughout

the purification process. Therefore, we cannot rule out that at least some of the newly found carotenoids were binding to the PSI monomer from bulk solution during the purification process. On the other hand, the clearly observed relatively high mobility of carotenoids, as demonstrated by their asymmetric distribution among the three monomers in the high-resolution crystal structure (2.5 Å) of the PSI trimer from *Synechocystis* sp. PCC6803 (Malavath *et al.* 2018), also opens up the possibility that carotenoid content in PSI is variable and perhaps dependent on the need for their light harvesting or photoprotective properties. The discrepancy between spectroscopically found decrease and structurally found increase in carotenoids could be explained by the circumstance that the total content of carotenoids, which is bulk solution carotenoids in the free detergent micelles and carotenoids bound to PSI, is not the same as the number of carotenoids displaying a density good enough for accurate modelling. Taking this into account, we hypothesized that bulk solution carotenoids bound to free detergent micelles are the main source of discrepancy between monomer and trimer absorption due to differing binding affinities to the hydrophobic interaction column. If true, then GraDeR prepared monomer and trimer PSI sample that almost completely lacks free detergent micelles, should display matching absorption spectra in the region of carotenoid absorption. This reasoning motivated us to re-measure the absorption spectra of GraDeR prepared PSI trimer and monomer sample. To our surprise, the discrepancy in absorption remained unchanged (see new Figure 2). Therefore, we have to conclude that the carotenoids responsible for enhanced absorption must be bound to the PSI trimer. The already described higher mobility of carotenoids compared to chlorophylls provides a rationale for a scenario where not all carotenoids bound to either monomeric or trimeric PSI can be found with defined density and modelled. All newly found carotenoids in our monomer structure are bound to chlorine rings located at the periphery of the complex. Surface exposed carotenoids bound to PSI only via hydrophobic interaction with chlorine rings should be particularly susceptible to loss by the detergent mediated membrane solubilization and subsequent purification steps. Given that the PSI monomer has a much greater area of detergent exposed surface in comparison to monomers in the trimeric complex, detergent induced loss of loosely bound surface exposed carotenoids that escape detection in X-ray and cryo-EM maps could well be increased. In this scenario some of peripheral bound carotenoids are giving a more defined density in our cryo-EM map, while relatively disordered peripheral carotenoids at the monomer-monomer interface are physically lost during the monomer purification resulting in the discrepancy of more modelled carotenoids versus loss of carotenoid absorption. Additionally, in order to quantify the mobility of caretonoids as well as subunits and chlorophylls in our structure, we employed a new method explained in detail in a recent paper of Pintilie *et al.* (Pintilie *et al.* "Measurement of atom resolvability in cryo-EM maps with Q-scores" *Nature Methods* 2020) and calculated Q scores for respective molecules. Q score define the resolvability of the molecules, providing a quantification of their flexibility. The results are shown in the new

Supplementary Table 3. According to this calculation, newly found carotenoids Q score and thus their resolvability is lower compared to other carotenoids and chlorophylls, indicating their relatively higher disorder.

These aspects are now discussed in the main text:

- Line 188-192: UV-Vis spectra of trimeric and monomeric PSI were measured at room temperature (Figure 2a) and difference spectra (trimer – monomer) were calculated after normalization at the chlorophyll a peak (678 nm) (Figure 2b). Positive peaks between 400 and 500 nm imply a higher amount of carotenoids in trimeric PSI.
- Line 300-302 : This finding might indicate the presence of unmodelled carotenoids in the trimer crystal structure.
- Line 312-314: BC-A853, had also been modelled in the very recently reported single particle cryo-EM structure of the *T. elongatus* PSI trimer.
- Line 469-472: This apparent higher mobility of carotenoids is reflected by their relatively low Q-score of 0.61 for all carotenoids and 0.53 for all newly found carotenoids in our model (see Supplementary Table 3 for Q-scores of all subunits, chlorophylls and carotenoids).
- Line 474-484: As we assume PsaK to be only disordered and not physically lost, the apparent clash of carotenoids with PsaK is likely avoided by an at least partial shift of this subunit upon monomerization. In support of a physiological role for this carotenoid, the newly found carotenoid BC-A853 was also observed in the cryo-EM structure of the *T. elongatus* PSI trimer. The UV-Vis difference spectrum (Fig. 2b) suggests that diminished absorbance in the carotenoid region for the monomer sample is the consequence loss from PSI bound carotenoid. Therefore, we posit that carotenoids at the monomer-monomer interface of PSI, which are shielded from solubilization in the trimer, are lost during the purification process.
- Line 736-738 : The absorption spectra were measured for GraDeR prepared PSI monomer and trimer using a UV-2450 spectrophotometer (SHIMADZU) in the range between 300 nm and 750 nm.

The new version of Figure 2 is shown below;

Figure 2: UV-Vis and 77 K spectra of trimeric and monomeric PSI from *T. elongatus*. a UV-Vis spectra. GraDeR prepared samples were diluted in buffer B + 0.005% (w/v) GDN. b UV-Vis difference spectra (trimer – monomer) were calculated after normalization at the chlorophyll a peak (678 nm). Differences related to the absorption of carotenoids (orange) and red chlorophylls (red) are indicated by the colored background. c 77 K Fluorescence spectra. Samples were diluted in buffer B + 0.03% (w/v) DDM. Measurements were carried out at an excitation wavelength of 436 nm with a step size of 1 nm and a bandpass filter of 5 nm.

The changes in Figure 2 are as follows;

- UV-Vis spectra are now showing the result of the new measurements. The difference between the previous and the new measurements is that, in the former sample, the free detergent micelles in the sample were not removed. The new measurements were done with the same samples after GRaDeR detergent

removal and thus, almost no free detergent micelles are present in the sample and thus presumably free pigments included in free micelles are removed.

- We added the difference spectra. Yellow shade shows the carotenoid related differences and red shade includes chlorophyll related differences between preparations of trimeric and monomeric PSI.

The new supplementary figure regarding Q scores is shown below;

Supplementary Table 3: Q-scores, a measure of local resolution by individual atom, for individual subunits, the total of modelled chlorophylls, the total of modelled carotenoids and newly modelled carotenoids.

Chain	Q-score	Estimated Resolution (Å)
A	0.71	2.30
B	0.70	2.37
C	0.69	2.41
D	0.68	2.50
E	0.64	2.69
F	0.48	3.54
I	0.65	2.62
J	0.52	3.40
K	-	-
L	0.64	2.68
M	0.66	2.59
X	-	-
Chlorophyll A	0.71	
Carotenoids	0.61	
Carotenoids new	0.53	

The major conclusion of the manuscript is that the loss of “red” chlorophylls that is well documented to happen with monomerization are not pigments that are adjacent to the trimeric interface, in particular those in proximity to the PsaL subunit that is involved in trimerization. This is an interesting and somewhat surprising finding.

Thank you very much for your interest in this finding.

I think the weakest part of the paper is the spectroscopy. The loss of red pigments is not well documented.

Though in the present study we did not include a precise estimation of the number of lost 'red' chlorophylls, we are in the fortunate situation that numerous measurements on this topic have been previously reported (Pålson *et al.* 1998, Gobets *et al.* 2001, Schlodder *et al.* 2014 among others). These previous studies agree that upon monomerization of the *T. elongatus* PSI trimer, two out of a total of four C719 "red" chlorophylls are lost. Since this spectroscopically observed loss of signal from C719 chlorophylls was found to be independent from the method of monomerization and also independent of the method of estimation, we deem that it can safely be assumed that also in the present study for each PSI complex two C719 are lost.

A careful difference spectrum between the monomeric and trimeric forms should be presented.

As indicated above, we have analyzed now the final samples after GraDeR purification by UV-Vis spectroscopy. We exchanged the UV-Vis absorption spectrum and added a difference spectrum to point out the differences between PSI monomers and trimers clearer (see new Figure 2 as shown above).

It is also not clear whether the "lost" chlorophylls are really lost or just disordered.

This is an interesting and very important point that has also been raised by reviewer #2 and which impacts the interpretation of our structural findings. Indeed, the question of whether 'red' chlorophylls that are lost upon monomerization of trimeric cyanobacterial PSI are physically lost or only disordered has been part of the discussion ever since the spectroscopy-based discovery of the dependence of 'red' chlorophyll absorbance and fluorescence emission on the oligomeric state of cyanobacterial PSI in the 1990's.

Our own cryo-EM based structural data can only show that no density was found that is equivalent to that of protein and chlorophylls, lipids or carotenoids previously found in the trimer crystal structure (PDB ID: 1jb0). Both physical loss and disorder among the 46,105 PSI monomer particles used for the 3D reconstruction are of equivalent explanatory power for the structural data. For the missing protein density, the partial presence of PsaF and the full presence of all known subunits in our 6.5 Å crystal structure, together with the detection of all subunits in our monomer preparation by SDS-PAGE and mass spectrometry, and our activity measurements give evidence favoring a monomerization induced disordering instead of physical loss. Our structurally found loss of 14 chlorophylls is not reflected by an equivalent loss of absorption, but rather a decrease in absorption confined to that of 'red' chlorophylls. This again favors disorder over physical loss. Previously the reversibility of monomerization induced loss of 'red' chlorophylls has been reported in situ as well as in vitro (Kruip *et al.* 1999; Domonkos *et al.* 2004). The in vitro study by Kruip *et al.* employed similar biochemical extraction and purification methods as in the present one. The previously reported recovery of 'red' chlorophyll absorption

and fluorescence through re-oligomerization of monomerized cyanobacterial PSI is also in support of disorder over physical loss.

However, the above described reasons do not rule out that in our cryo-EM density map detected loss of density is the consequence of physical loss induced by the single particle cryo-EM sample preparation. Indeed, the adherence of the vast majority of particles to the air water interface in the time between thinning of the liquid film over holey carbon by either blotting or self-wicking nano-wire grids and vitrification via plunge freezing is now well documented (Noble *et al.* eLife, 2018). That the interaction of protein particles with the air water interface can be both harmful and leading to preferred particle orientations has likewise been demonstrated (Noble *et al.*, eLife, 2018; Noble *et al.*, Nature Methods, 2018). However, the crispiness of the PsaL density in our map is in conflict with the idea of a general sensitivity of *T. elongatus* PSI complexes to single particle cryo-EM vitrification. More importantly, in the newly available *T. elongatus* PSI trimer single particle cryo-EM density map (PDB ID: 6tra; EMD ID: EMD-10557) all the protein and chlorophyll missing in our map are present and fully modelled. In other words, the strong density for chlorophyll B1233, chlorophyll B1219, the loop region B291-316, lipid IV and PsaX of the trimer single particle cryo-EM structure gives evidence that the vitrification process itself is very likely not causing the observed disorder in our single particle cryo-EM monomer structure. We therefore believe, after much discussion and thought on this important question, that the absence of density for both protein and chlorophyll and the loss of 'red' chlorophylls is originating from monomerization induced disorder and not physical loss. This conclusion is now reflected in the revised manuscript and impacts the possible biological meaning of cyanobacterial PSI oligomeric state. Reversible monomerization induced disorder in the 'red' chlorophyll cluster area adjacent to PsaX could function as a switch for C719 activity that is controlled by the oligomeric state and thus serving cyanobacterial adaption to changing light conditions. Given that 'red' chlorophylls make up only a small part of the total chlorophyll content and can serve in both light harvesting and excess energy dissipation, oligomerization as a molecular control mechanism does make biological sense. Furthermore, the "outsourcing" of 'red' chlorophylls to outer light harvesting complexes in algae and plants makes control of 'red' chlorophyll activity by oligomeric state redundant and as a consequence presents a reasonable explanation for the absence of PSI oligomers in algae and plants:

- Line 383-385: However, the possibility of a partial removal of subunits or co-factors as a result of the LDAO treatment cannot be completely ruled out.
- Line 401-417: Recently, the long-suspected notion that in single particle cryo-EM experiments the vast majority of protein complexes adheres to the air-water interface which can result in preferred orientations and harm the protein complex in question has been conclusively demonstrated. This specific limitation of single particle cryo-EM based determination of protein complex structures potentially also applies to the regions of disorder observed for the *T. elongatus*

PSI monomer. However, the same studies also showed that these air-water interface effects are highly protein dependent and cannot be generalized. Therefore, for the present study it is highly informative that in the now reported single particle cryo-EM structure of the *T. elongatus* PSI trimer (PDB ID 6TRA, EMD ID EMD-10557) all subunits and co-factors that were found in the original X-ray crystal structure (PDB ID 1JB0) could be modelled. Moreover, the 6TRA model of trimeric *T. elongatus* PSI also included an improved model of PsaK. Importantly, it also visualized all chlorophylls and the protein regions of the here identified PsaX region of disorder in the monomer.

- Line 429-435: However, the reported reversibility of the process points to an oligomeric state dependent conformational disorder that could function as a molecular switch for C719 activity (red chlorophyll activity) and thus as a cyanobacterial response to changing light conditions. With 'red' chlorophylls of PSI in algae and plants being located in external antenna light harvesting complexes, oligomerization as a molecular mechanism for controlling their activity becomes redundant.

It should be an easy job to measure the total number of pigments in the preparation by extraction of a sample and take the spectrum of the extracted pigment and then measure the P700 content by either light-induced or chemically induced spectroscopy. Using the known extinction coefficients of chlorophyll and P700 will give the number of chlorophylls per P700. Carotenoids can be similarly measured. I think this needs to be done to put the whole question of the number of lost pigments on a firm footing.

We have collaborated intensively with Navassard Karapetyan in the past for in depth spectroscopic analysis of isolated PS1 complexes (e.g. El-Mohsnawy et al. 2010, *Biochemistry*, 49,4740-4751). Unfortunately, Prof. Karapetyan passed away and our laboratories do not have comparable instrumentation or experience. Particularly, we don't have access to deep-temperature UV-VIS or pump-probe spectroscopy for detailed analysis. However, we tried to measure chemical and light induced P700 difference spectra at room temperature:

The chemical induced difference spectra are very noisy with a quite high standard-deviation (indicated by dashed lines) and the determined Chl/P700 ratio differ substantially from the expected value of ~96, although we tried to minimize any handling error. The flash-induced difference spectra show as expected a clear negative peak at 702 nm with good reproducibility. Albeit, the determined Chl/P700 ratios are far away from reasonable values. Most probably, we miss the fully oxidized P700 state due to the insufficient time resolution of the instrument. We understand that the pigment composition of the samples is an interesting aspect that should be investigated but based on our preliminary experiments we concluded that it is not as easy as expected and we would need specialized expertise for proper measurements. However, it is not essential to know the specific pigment ratios for the conclusions drawn in the paper and a detailed analysis would go beyond the scope of this work.

Reviewer #2 (Remarks to the Author):

The manuscript by Coruh et al. reports the structural analysis of the monomeric PSI complex from *T. elongatus*. Monomeric PSI is characterized by the loss of red chlorophylls, which are present in the trimer. Several candidates of the red chl clusters were suggested, including chlorophyll clusters at the monomer-monomer

interface. However, no conclusive assignment could be made till now. In this work, the authors solved structures of monomeric PSI, using the cryo-EM method (3.2 Å) and x-ray crystallographic method (6.5 Å). Compared with the structure of trimeric PSI from the same strain, They found: 1) PsaK and PsaX could not be modeled in their cryo-EM structure, and PsaF and PsaJ were only built to about a fifth of the trimer crystal structure. However, the electron density of the transmembrane helices of these subunits were observed in the low-resolution crystal structure. 2) the monomer structure contains eight carotenoids located at novel sites, which are absent in the trimer, while four carotenoids in the trimer are absent in the monomer. 3) the monomer contains 14 less chlorophylls compared with the trimer, therefore some of these chlorophylls might be responsible for the red spectra of the PSI trimer. The authors suggested a cluster of chlorophylls adjacent to PsaX contributes to the red spectra of PSI trimer.

Overall, this is an interesting contribution.

Thank you for finding our study of interest.

However, there are some points that I would like the authors to address.

- Previously, a structure of monomeric PSI from *Synechocystis* 6803 was reported, it is encouraged that the authors add several sentences describing the comparison of the two monomer structures.

Thank you very much for this valuable suggestion. We find that further comparison with PSI *Synechocystis* sp., which is the only other PSI of which both trimer and monomer structures are available, is indeed very interesting especially in the region of the PsaX chlorophyll cluster. The circumstance that *Synechocystis* PSI lacks C719 'red' chlorophylls and also lacks PsaX and the luminal loop extension of PsaB that is anchoring B1233 has been recognized even before the arrival of the first structure of a *Synechocystis* PSI and employed to argue that B1233 can be assigned as a C719 chlorophyll (Karapetyan *et al.* 2006). That we find B1233 to be disordered in our monomer structure while the other two 'staircase' chlorophylls B1231 and B1232 are conserved and visualized in both *Synechocystis* trimer and monomer structure is in strong support of this notion and we assign B1233 to be one of the lost 'red' C719 chlorophylls. Further comparison on the stromal side revealed that *Synechocystis* PSI lacks the short loop-helix-loop region with its two phenylalanine sidechains that supports the loose coordination of B1219, but instead contains an additional third chlorophyll that if present in *T. elongatus* PSI would clash with PsaX. It has previously been stated that excitonic coupling alone cannot be responsible for the full red shift of C-719 chlorophylls, but that water coordination of the chlorine ring magnesium likely is as important (Karapetyan *et al.* 2006). Notably, both B1233 and B1219 have water coordinated magnesium. Thus, it appears likely that B1219 is the second lost C719. Moreover, while our manuscript is in revision, the Mazor lab has reported the spectral properties and structure of a *Synechocystis* PCC 6803 chimera with an luminal loop extension matching that of *T. elongatus* PS1 which is supporting the putative 'red' chlorophyll B1233 (Toporik *et al.*, Nat. Comm., 2020). They find that the chimeric PSI not only has a chlorophyll bound in the expected

B1233 position, but also that the chimeric PSI is now red shifted relative to the native *Synechocystis* PCC 6803 PS1. Their findings are in strong support of our assignment of B1233 as a 'red' chlorophyll and underline the importance of this region of the PS1 complex as the site for the elusive 'red' chlorophyll cluster. The above described comparison of *T. elongatus* PSI with *Synechocystis* PSI is reflected in the revised manuscript by an additional section in our discussion and a new Supplementary Figure 7.

- LINE 558-593: PSI from the mesophilic cyanobacterium *Synechocystis* sp. is the only cyanobacterial photosystem I for which both trimer and monomer crystal structures have already been reported. Notably, in contrast to *T. elongatus* PSI, *Synechocystis* sp. PSI does not contain C719 'red' chlorophylls. This prompted us to compare the region of lost chlorophylls adjacent to PsaX from *T. elongatus* with the same region of the PSI complex of *Synechocystis* sp. in both trimer and monomer structures (Supplementary Figure 7). Important differences are the absence of PsaX and the absence of B1233 with its supporting loop region of B490-496 in *Synechocystis* PSI. Interestingly, these differences had been noticed even before the arrival of the first *Synechocystis* crystal structure. Moreover, the loop formed by the residues B306-316 of the loop-helix-loop region B291-316 in the *T. elongatus* PSI trimer structure, that with its two coordinating phenylalanine side chains is stabilizing the very weakly bound B1219, disordered in our monomer structure, is also missing in *Synechocystis* sp. PSI. Instead *Synechocystis* sp. PSI has an additional third chlorophyll which, if present in PSI of *T. elongatus*, would clash with PsaX.

Significant structural differences of *T. elongatus* PSI with *Synechocystis* sp. PSI are concentrated in the main region of disorder in monomerized *T. elongatus* PSI. And the most significant spectroscopic difference between the two species lies with the absence of C719 chlorophylls in *Synechocystis* PSI. This provides strong support for the assignment of B1233 and B1219 as C719 'red' chlorophylls. For strong red shifts in chlorophylls excitonic coupling alone is thought to be insufficient and the importance of magnesium coordination by water has been suggested as a factor allowing stronger red shifts. We find that the absence of the water coordinated 'outer staircase' chlorophyll B1233 and the replacement of water coordination by histidine coordination in B1219 of *Synechocystis* PSI is

in support of this notion (see also Supplementary Figure 7). In a very recent study a chimeric *Synechocystis* PCC 6803 PSI into which the outer staircase chlorophyll B1233 supporting luminal PsaB loop was introduced has been characterized by both cryo-EM and spectroscopy. The cryo-EM based atomic model showed that an additional chlorophyll corresponding to B1233 is present in the chimeric PSI complex and that it is red shifted. These findings are in strong support for our assignment of B1233 as a 'red' chlorophyll and of the PsaX region as the site of the 'red' chlorophyll cluster in *T. elongatus* PSI.

In the new Supplementary Figure 7 structural differences between the PsaX region of *T. elongatus* PSI and the corresponding region of *Synechocystis* sp. PSI are illustrated as shown below;

Comparison of *T. elongatus* with *Synechocystis* sp. PCC 6803

Structural differences between in PSI	
T. elongatus and Synechocystis sp. at 'red' chlorophyll cluster region	
Between trimer structures	Between monomer structures
Stromal loop (B306-B316) above B1219 in T. elongatus	Disorder of B1218 and 1219 amongst other chlorophylls in T. elongatus
Chlorophyll B1233 to form a trimer with B1232 and B1231 in T. elongatus	Absence of B1240 in T. elongatus
Luminal loop (B490-B499) below B1233 in T. elongatus	Presence of lipid B5004 in T. elongatus
Chlorophyll B1240 forming a trimer with B1218 and B1219 in Synechocystis sp.	Absence of a loop (B490-B499) in Synechococcus sp.

Supplementary Figure 7: Structural differences between *T. elongatus* and *Synechocystis* sp. PCC 6803 PSI in the 'red' chlorophyll cluster region on the periphery of PsaB.

- In order to obtain the monomeric PSI, the authors solubilized thylakoid membrane with LDAO, and purified the PSI monomer through HIC. The harsh treatment may result in the loss of the peripheral subunits, such as PsaF, PsaJ, PsaK, and PsaX. This possibility should be stated in the manuscript.

This possibility is now stated in the manuscript;

- Line 383-385: However, the possibility of a partial removal of subunits or co-factors as a result of the LDAO treatment cannot be completely ruled out.

- Line 390-392, It is obvious that PsaF, PsaJ, PsaK, and PsaX are not physically lost during the purification, however, it is possible that they are dissociated from other core subunits during the grid preparation, which happens frequently in the cryo-EM sample preparations.

This important point is now discussed in the manuscript and the relevant literature cited as described in our response to reviewer #1;

- Line 401-417: Recently, the long-suspected notion that in single particle cryo-EM experiments the vast majority of protein complexes adheres to the air-water interface which can result in preferred orientations and harm the protein complex in question has been conclusively demonstrated. This specific limitation of single particle cryo-EM based determination of protein complex structures potentially also applies to the regions of disorder observed for the *T. elongatus* PSI monomer. However, the same studies also showed that these air-water interface effects are highly protein dependent and cannot be generalized. Therefore, for the present study it is highly informative that in the now reported single particle cryo-EM structure of the *T. elongatus* PSI trimer (PDB ID 6TRA, EMDB ID EMD-10557) all subunits and co-factors that were found in the original X-ray crystal structure (PDB ID 1JB0) could be modelled. Moreover, the 6TRA model of trimeric *T. elongatus* PSI also included an improved model of PsaK. Importantly, it also visualized all chlorophylls and the protein regions of the here identified PsaX region of disorder in the monomer.

- Based on the text (Line 310-315, 429-432) and Supplementary Fig. 5, the newly found carotenoids (852, 855 and 856) seem to occupy the site of PsaK. However, the authors clearly explained that PsaK is present in the complex. Therefore it is difficult to imagine how these carotenoids bind at the PsaK site.

That we found density for a carotenoid in our monomer cryo-EM map which is clashing with the position of PsaK of the trimer crystal structure is surprising. Since we believe PsaK to be not physically lost from the monomeric complex, neither during purification nor cryo-EM sample preparation this should indicate that PsaK is not only disordered, but also at least slightly dislocated by the monomerization. This unexplained contradiction is now discussed in the revised manuscript;

- Line 474-479 : As we assume PsaK to be only disordered and not physically lost, the apparent clash of carotenoids with PsaK is likely avoided by an at least partial shift of this subunit upon monomerization. In support of a physiological role for this carotenoid, the newly found carotenoid BC-A853 was also observed in the cryo-EM structure of the *T. elongatus* PSI trimer.

- The spectroscopic results indicated a decrease of carotenoids in the monomer versus the trimer, while the structural analysis showed larger number of carotenoids modeled in the monomer. Is there any explanation for the contradictory between the spectroscopic and structural results?

This point has also been raised by reviewer #1 and discussed in detail in our response. Additional absorption measurements on free detergent micelles less GraDeR sample are now described in the main text and Supplementary Figure 8;

- Line 188-192: UV-Vis spectra of trimeric and monomeric PSI were measured at room temperature (Figure 2a) and difference spectra (trimer – monomer) were calculated after normalization at the chlorophyll a peak (678 nm) (Figure 2b). Positive peaks between 400 and 500 nm imply a higher amount of carotenoids in trimeric PSI.
- Line 300-302 : This finding might indicate the presence of unmodelled carotenoids in the trimer crystal structure.
- Line 312-314: BC-A853, had also been modelled in the very recently reported single particle cryo-EM structure of the *T. elongatus* PSI trimer.
- Line 474-484: As we assume PsaK to be only disordered and not physically lost, the apparent clash of carotenoids with PsaK is likely avoided by an at least partial

shift of this subunit upon monomerization. In support of a physiological role for this carotenoid, the newly found carotenoid BC-A853 was also observed in the cryo-EM structure of the *T. elongatus* PSI trimer. The UV-Vis difference spectrum (Fig. 2b) suggests that diminished absorbance in the carotenoid region for the monomer sample is the consequence loss from PSI bound carotenoid. Therefore, we posit that carotenoids at the monomer-monomer interface of PSI, which are shielded from solubilization in the trimer, are lost during the purification process.

- Line 736-738 : The absorption spectra were measured for GraDeR prepared PSI monomer and trimer using a UV-2450 spectrophotometer (SHIMADZU) in the range between 300 nm and 750 nm.

Minor points:

- The authors identified eight more carotenoids in the monomeric structure compared with the trimeric structure. It's better to show the cryo-EM densities of these carotenoids.

Thank you very much for this suggestion. Cryo-EM density maps and the corresponding models of all eight additional carotenoids of the monomeric structure are now shown in the new Supplementary Figure 6.

- Please provide the details PSI activity (Line 174-176) measurement in the METHODS.

We are grateful for reviewer #2 pointing out this blunder in our manuscript. The missing method section for the activity measurement is now included in the revised manuscript;

- Line 684-697: PSI activity was indirectly measured using the Mehler reaction as previously described via the time-dependent oxygen consumption, detected with an optically-based oxygen sensor system (PST3, FIBOX 3 (PreSens™)). Monomeric or trimeric PSI were added to 1 ml of activity buffer (25 mM HEPES (pH 7.5), 500 mM NaCl, 3 mM MgCl₂, 330 mM Mannitol) at a final concentration of 5 µg Chl/ml, supplied with 0.8 mM dichloro-phenol-indophenol (DCPIP) and 5 mM sodium ascorbate as sacrificial electron donors and 1.2 mM methyl viologen as electron acceptor. The sample mixture was prepared inside a tempered, opaque cuvette at 30 °C under constant stirring. After an initial incubation of 2 min in the dark the measurement was started by switching on

the measuring light ($\lambda=680$ nm) for 3 min. The PSI-dependent oxygen consumption was calculated via the linear slope and specified as:

- Line 315, Supplementary Figure 4 should be Supplementary Figure 5.

Thank you for spotting this typo which is now corrected.

- Line 119, ref. 22 reports the tetrameric PSI from Chroococidiopsis sp TS-821, not from Anabaena 7120.

Again, we thank reviewer #2 for pointing out this error, which is now corrected.

REVIEWERS' COMMENTS:

Reviewer #1 (Remarks to the Author):

The revised manuscript is acceptable for publication.

Reviewer #2 (Remarks to the Author):

My questions were carefully answered, and the manuscript has been much improved. I have no major issue to the manuscript.

Minor points:

-It's still difficult to understand how these newly modeled carotenoids at monomer-monomer interface show more defined density in monomers than in trimers. From Supplementary Fig. 6, it seems that most of these carotenoids were only partially modeled, which indicates that they are not highly ordered. Did the authors consider the possibility that the densities assigned as carotenoids are corresponding to some unmodeled residues or lipids in the molecule?

-The fluorescence spectra (Fig. 2) showed that the monomer is red-shifted compared with the trimer, why? Is the label wrong?

Point-by-point response:

Reviewer #1 (Remarks to the Author):

The revised manuscript is acceptable for publication.

Thank you very much for your acceptance of our manuscript.

Reviewer #2 (Remarks to the Author):

My questions were carefully answered, and the manuscript has been much improved. I have no major issue to the manuscript.

Thank you very much for your positive comments.

Minor points:

-It's still difficult to understand how these newly modeled carotenoids at monomer-monomer interface show more defined density in monomers than in trimers. From Supplementary Fig. 6, it seems that most of these carotenoids were only partially modeled, which indicates that they are not highly ordered. Did the authors consider the possibility that the densities assigned as carotenoids are corresponding to some unmodeled residues or lipids in the molecule?

Since we did not observe any branching of the density and for example B844 did actually exhibit some possible methyl bumps, we believe it most likely that it is carotenoids that are underlying these densities.

-The fluorescence spectra (Fig. 2) showed that the monomer is red-shifted compared with the trimer, why? Is the label wrong?

This indeed was a sloppy labeling mistake on our side that is now corrected. We recognized this after uploading the files, but the fact that Reviewer #2 also spotted this mistake gives us confidence that no others are left. We apologize for this and thank you very much again for carefully checking our manuscript.